# Similarity-based transfer learning with deep learning networks for accurate CRISPR-Cas9 off-target prediction

Jeremy Charlier[1]☉, Zeinab Sherkatghanad[1]☉, Vladimir Makarenkov (iD)[1,2]*

**1** Département d'Informatique, Université du Québec à Montréal, Montreal, Quebec, Canada,
**2** Mila - Quebec AI Institute, Montreal, Quebec, Canada

☉ These authors contributed equally to this work.
* makarenkov.vladimir@uqam.ca

**Data availability statement:** All relevant data are within the manuscript and its Supporting information files. The submission contains all raw data required to replicate the results of our study.

## Abstract

Transfer learning has emerged as a powerful tool for enhancing predictive accuracy in complex tasks, particularly in scenarios where data is limited or imbalanced. This study explores the use of similarity-based pre-evaluation as a methodology to identify optimal source datasets for transfer learning, addressing the dual challenge of efficient source-target dataset pairing and off-target prediction in CRISPR-Cas9, while existing transfer learning applications in the field of gene editing often lack a principled method for source dataset selection. We use cosine, Euclidean, and Manhattan distances to evaluate similarity between the source and target datasets used in our transfer learning experiments. Four deep learning network architectures, i.e. Multilayer Perceptron (MLP), Convolutional Neural Networks (CNNs), Feedforward Neural Networks (FNNs), and Recurrent Neural Networks (RNNs), and two traditional machine learning models, i.e. Logistic Regression (LR) and Random Forest (RF), were tested and compared in our simulations. The results suggest that similarity scores are reliable indicators for pre-selecting source datasets in CRISPR-Cas9 transfer learning experiments, with cosine distance proving to be a more effective dataset comparison metric than either Euclidean or Manhattan distances. An RNN-GRU, a 5-layer FNN, and two MLP variants provided the best overall prediction results in our simulations. By integrating similarity-based source pre-selection with machine learning outcomes, we propose a dual-layered framework that not only streamlines the transfer learning process but also significantly improves off-target prediction accuracy. The code and data used in this study are freely available at: https://github.com/dagrate/transferlearning_offtargets.

## Author summary

CRISPR-Cas9 is a popular gene-editing technology that allows researchers to modify an organism's genomic DNA at precise locations. Significant research efforts have

**Funding:** This work was supported by Fonds de recherche du Québec – Nature et technologies (Grant 173878 to VM) and Natural Sciences and Engineering Research Council of Canada (Grant 249644 to VM). The funders had no role in study design, data collection and analysis, decision to publish, or preparation of the manuscript.

**Competing interests:** The authors have declared that no competing interests exist.

been focusing on improving its precision and effectiveness, with particular emphasis on minimizing off-target effects. At the same time, transfer learning techniques are becoming increasingly important for addressing deep learning challenges in computational biology, especially in the field of CRISPR-Cas9, where plausible training data availability can be limited. This study investigates the effectiveness of integrating similarity-based analysis with transfer learning for improving CRISPR-Cas9 off-target prediction. Our key contribution consists in an experimental evaluation of three distance metrics, i.e. cosine, Euclidean, and Manhattan distances, along with several traditional machine learning and deep learning models, in the context of knowledge transfer by transfer learning applied to gene editing data. For each considered target dataset our transfer learning framework determines the most suitable source dataset to be used in the model pre-training. The proposed computational framework offers a reliable and systematic method for selecting suitable source data, streamlining the transfer learning process, and improving prediction accuracy.

## Introduction

CRISPR-Cas9 (Clustered Regularly Interspaced Short Palindromic Repeats and the associated protein 9) has become a leading technology for precise and efficient genome (or gene) editing, allowing genetic material to be added, removed, or altered at particular locations of a given genome. Its simplicity, high precision, and versatility across various applications have made it a dominant tool in the field [1–4]. The CRISPR-Cas9 genetic engineering system reflects the immune defense mechanism of certain bacteria. Bacteria identify the invading viral DNA and cut out a segment of the virus DNA, known as a protospacer, to insert it into the front of the CRISPR array. Bacteria are armed by the protein Cas9 to produce RNA segments from CRISPR arrays to cut the DNA of the phage virus, and thus defend themselves from the phage infection return [5]. In CRISPR-Cas9, single-guide RNA (sgRNA) consists of a crRNA and tracrRNA duplex that guides Cas9 to its Protospacer-Adjacent Motif (PAM) target at the end of the DNA sequence [6]. The PAM sequence that follows the protospacer sequence in a viral genome helps Cas9 to distinguish between itself and the enemy. The CRISPR-Cas9 gene editing system covers many areas of human health and welfare [7,8]. The technology has demonstrated important clinical potential for drug development to treat various human diseases, including cancer [9–11], for preventing genetic disorders in plant genetic engineering [12–14], for providing animal disease treatment [15,16], as well as for assisting bio-fuel production [17,18].

A significant challenge in the CRISPR-Cas9 gene editing process is the off-target effect, where the sgRNA targets DNA fragments other than the original DNA fragment aimed, resulting in unwanted cuttings of the DNA sequence [19,20]. To ensure safe, reliable, and efficient application of the CRISPR-Cas9 technology, it is essential to develop an accurate method to maximize the on-target efficiency and minimize the number of potential off-targets. There are common scoring methods for off-target prediction, such as CFD score [21], MIT score [22], CHOPCHOP [23] and CCTop score [24] that are based on specific scoring function highlighting mismatch locations. The main disadvantage of the traditional scoring methods is their incapability to improve predictive performance when the number of samples increases as well as their inability to discover relationships between mismatched and matched sites [25]. Nowadays, effective and feasible solutions to address these issues are provided by data-driven algorithms [26]. The modern data-driven models that rely on deep learning

(DL) show promising results with the growing number of CRISPR-Cas9 data; they typically outperform existing scoring methods in terms of off-target prediction [25].

However, deep learning models employ thousands of parameters, requiring a substantial number of samples in CRISPR-Cas9 datasets. To this end, Transfer Learning (TL) has emerged as an effective approach to overcome the problem of insufficient number of samples [25,27,28]. TL is used to learn properties of large source datasets in order to transfer them to smaller target datasets. TL is employed to improve the prediction accuracy and to avoid data overfitting on small datasets by leveraging the knowledge learned from larger datasets having similar properties.

Although some CRISPR-Cas9 benchmark datasets for on- and off-target prediction are currently available [22], the number of samples they contain is often insufficient to achieve accurate deep learning predictions. In this case, TL can be viewed as a viable alternative approach to the use of traditional machine learning (ML) or more sophisticated DL models which are both prone to overfitting when the data availability is limited. Recently, Lin et al. [25] and Charlier et al. [28] used TL to predict off-targets in small CRISPR-Cas9 datasets. Precisely, they trained the model on a large CRISPOR dataset (18,236 samples) to predict off-targets in a much smaller GUIDE-Seq dataset (430 samples). Elkayam et al. [29] introduced the DeepCRISTL model, pretraining it on high-throughput source datasets, including more than 150 000 gRNAs. Then, using TL, they successfully applied DeepCRISTL on target data consisting of much smaller functional or endogenous datasets. Zhang et al. [30] proposed the C-RNNCrispr model to predict sgRNA activity using convolutional and recurrent neural networks (CNNs and RNNs, respectively). After pretraining their model on benchmark data, the authors applied TL by using small-size datasets to fine-tune C-RNNCrispr. Zhang et al. [31] developed two attention-based CNN models, called CRISPR-ONT and CRISPR-OFFT, for on- and off-target prediction, respectively. They employed TL for small-size cell-line sgRNA specificity prediction. Zhang et al. [32] applied TL by using their pre-trained Hybrid CNN-SVR model that was fine-tuned to provide predictions for small sample cell-line datasets. Yaish et al. [33] also proposed a novel DL network leveraging TL for off-target activity prediction. The authors introduced some innovative metrics and visualization techniques to enhance the understanding of the bulges impact on genome editing. Elkayam et al. [34] developed the DeepCRISTL model to predict the editing efficiency in a specific cellular context. The authors proposed and compared four TL approaches that are as follows: (a) the full approach that fine-tunes all model weights; (b) the last-layer approach that fine-tunes only the weights of the last hidden layer and of the output layers; (c) the no-embedding/convolution approach that fine-tunes all model weights besides those of the embedding and the convolutional layers; (d) the gradual-learning approach that first fine-tunes the weights of the last hidden and the output layers, and then all other model weights with a smaller learning rate.

We need to point out that different CRISPR-Cas9 datasets are collected under distinct laboratory conditions and equipment, resulting in different data patterns and distributions. The prediction accuracy provided by the TL technique under consideration for a given target dataset depends drastically on the similarity between this target dataset and the source dataset used in pretraining. In other words, using completely different source and target datasets in TL should not lead to satisfactory prediction results for the target data. This paper addresses the challenge of similarity of the source and target datasets used in transfer learning experiments with CRISPR-Cas9 off-target data.

Our key contributions are outlined below:

- First, we propose a robust dual-layer framework that integrates similarity-based pre-evaluation with transfer learning for off-target predictions in CRISPR-Cas9 (see Fig 1).

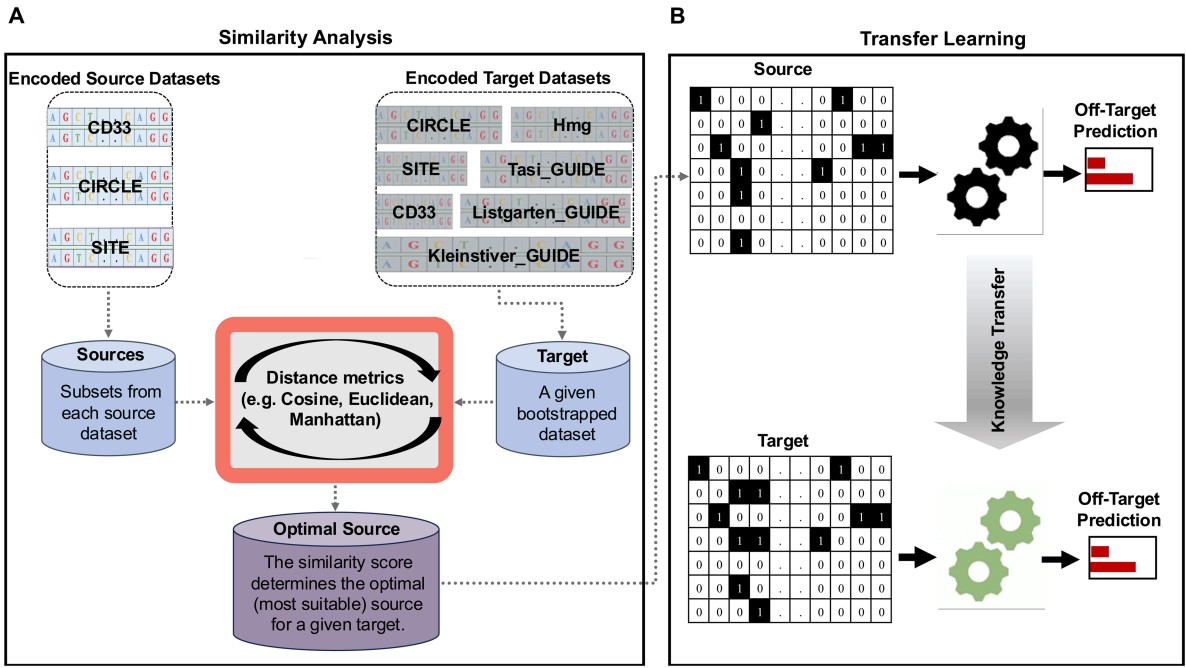

**Fig 1. An overview of the proposed framework leveraging data similarity analysis with genome editing transfer learning.** (A) Three distance measures: cosine, Euclidean, and Manhattan distances are used to identify the most suitable source dataset, among three benchmark candidate datasets CD33, CIRCLE, and SITE (complete large dataset), for a given target dataset (smaller bootstrapped dataset); (B) The framework subsequently transfers the learned model knowledge from the selected optimal source dataset to the target dataset, enhancing the predictive accuracy.

In contrast to previous studies applying transfer learning to CRISPR-Cas9 datasets, our approach first compares the sgRNA-DNA sequence patterns of the source and target datasets using cosine, Euclidean, or Manhattan distance to identify the optimal source-target pair. The model knowledge is then transferred from a source dataset with a similar sgRNA-DNA sequence pattern to the target dataset.

- Second, we compare the suitability of cosine, Euclidean, and Manhattan distances for transfer learning experiments in CRISPR-Cas9, based on the performance of data-driven DL and ML models.
- Third, we identify the best-performing DL and ML models using reliable performance evaluation metrics to effectively predict off-targets in CRISPR-Cas9.
- Fourth, we demonstrate the effectiveness of our proposed framework by applying it to the analysis of seven popular benchmark datasets: CD33, CIRCLE, SITE, Tasi_GUIDE, Listgarten_GUIDE, Kleinstiver_GUIDE, and Listgarten_Elevation_Hmg.

We need to highlight that existing transfer learning applications in CRISPR-Cas9 (e.g. DeepCRISTL [34], C-RNNCrispr [31], CRISPR-ONT and CRISPR-OFFT [31]) often lack a principled method for effective source dataset selection. Thus, our key contribution is in optimizing the transfer learning process through intelligent source selection, and not in inventing new deep learning architectures designed for transfer learning experiments.

## 1. Materials and methods

### 1.1. Datasets

We conducted our off-target prediction experiments on seven well-known public CRISPR-Cas9 datasets:

- **CD33 dataset** was constructed and made available by Doench et al. [35]. It consists of gRNA-target pairs with only mismatches, comprising 4,853 gRNAs targeting the human coding sequence of CD33. This is one of the rare well-balanced datasets in which the class imbalance ratio is 0.81 (i.e. close to 1).
- **CIRCLE dataset** contains gRNA-target pairs with both mismatches and indels from 10 different guide-RNAs. In our study, we voluntarily modified the encoding process to aggregate mismatches and indels into the same group. This was a deliberate choice on our end in order not to bias the results of our experiments. The dataset contains 7,371 active off-targets, which were validated using the Circularization for In vitro Reporting of CLeavage Effects by sequencing (CIRCLE-seq) technique [36]. Additionally, Lin et al. [37] used Cas-Offinder [38] to identify 577,578 inactive off-target genomic sites in this dataset, including mismatches and indels.
- **SITE dataset** contains 217,733 sgRNA-DNA sequence pairs with 9 guide sequences; 3,767 of them correspond to active off-targets. The dataset is validated by the SITE-Seq [39,40] biochemical method which employs Cas9 programmed with sgRNAs to recognize cut sites within genomic DNA.
- **Tasi_GUIDE dataset** has been provided by Tsai et al. [41] based on the cellular method, called GUIDE-seq. This dataset includes a total of 294,534 target sites, with 354 off-target sites containing mismatches.
- **Listgarten_GUIDE** is the fifth dataset used in our experiments, comprising 56 minority class samples and 383,463 majority class samples, validated with the GUIDE-seq technology [42].
- **Kleinstiver_GUIDE** dataset consists of 54 positive off-target sites and 95,775 inactive off-target sites, validated by the GUIDE-seq technology [43].
- **Listgarten_Elevation_Hmg** dataset, referred to as Hmg, comprises 52 active off-targets among 10,129 potential off-target sites from 19 gRNAs, which was organized and made publicly available by Haeussler et al. [22].

The dataset's name, the CRISPR-Cas9 technique used, the number of gRNAs, the number of samples in both the minority and majority classes, and the class imbalance ratio for each of these datasets are summarized in Table 1. The datasets are publicly available in our GitHub repository at: https://github.com/dagrate/transferlearning_offtargets.

**Table 1. Seven CRISPR-Cas9 benchmark off-target datasets used in our study.** Six of them include gRNA-target pairs with mismatches only, and one of them (CIRCLE, denoted with an asterisk) includes gRNA-target pairs with both mismatches and indels. Minority class samples correspond to active off-target sites (or active off-targets) and Majority class samples correspond to inactive off-target sites.

| Dataset | CRISPR-Cas9 technique | gRNAs | Minority class samples | Majority class samples | Class imbalance ratio |
|---|---|---|---|---|---|
| CD33 | Protein Knockout Detection | 65 | 2,273 | 2,580 | 0.8810 |
| CIRCLE* | CIRCLE-Seq | 10 | 7,371 | 577,578 | 0.0128 |
| SITE | SITE-Seq | 9 | 3,767 | 213,966 | 0.0176 |
| Tasi_GUIDE | GUIDE-Seq | 9 | 354 | 294,180 | 0.0012 |
| Listgarten_GUIDE | GUIDE-Seq | 22 | 56 | 383,463 | 0.0001 |
| Kleinstiver_GUIDE | GUIDE-Seq | 5 | 54 | 95,775 | 0.0005 |
| Listgarten_Elevation_Hmg | PCR, Digenome-Seq and HTGTS | 19 | 52 | 10,077 | 0.0052 |

## 1.2. Data encoding

To encode sgRNA-DNA sequences, we adopted the encoded scheme introduced by Lin et al. [37] that integrates mismatches, insertions, deletions, and matches to preserve the mutual information between on-target and off-target sites. This scheme represents each sgRNA-DNA sequence pair using seven-bit one-hot encoding: a five-bit channel (A, C, G, T, _) and a two-bit direction channel used to indicate the insertion/indel or mismatch directions. Consequently, a $7 \times 23$ matrix (where 23 represents the sequence length, including the 3-bp PAM adjacent to the 20 bases) allows for considering three types of base mismatches, missing bases (RNA bulge or insertion), and extra bases (DNA bulge or deletion) in off-target sites. An overview of this encoding technique, with examples including an insertion (RNA bulge), a mismatch, and a deletion (DNA bulge) is presented in Fig 2.

## 1.3. Data splitting procedure for model training

We specifically selected three datasets, CD33, CIRCLE, and SITE, as potential source datasets due to a large number of positive samples in their minority class (i.e. active off-targets) and the lowest class imbalance among the seven datasets considered, as indicated in Table 1. This selection enhances the robustness of our analyses during the training process.

We used a standard train-test split from the scikit-learn [44] implementation with shuffling, a ratio of 0.3, and equal stratification of the classes. The stratification was employed to ensure that the class distribution within the training and testing datasets accurately reflects the original class proportions before the train-test split. By maintaining relative class ratios, stratification mitigates biases and enhances the reliability of model evaluation to address the issue of data imbalance [45].

## 1.4. Model description

An overview of the classification models used in our experiments is provided in Supporting Information. The two following Python libraries were used for model implementation:

- **Scikit-Learn ML and DL models:** Four classification models were implemented using this library: One Hidden Layer Perceptron (MLP1), Two Hidden Layer Perceptron (MLP2), Random Forest (RF) classifier, and Logistic Regression (LR) classifier. These models are well-established ML and DL techniques widely used in practical applications.
- **DL networks with TensorFlow:** In Supporting Information, we provide details on eight deep neural network models implemented using the Python package TensorFlow. They include a three-layer feedforward neural network (FNN3), a five-layer FNN (FNN5), and a ten-layer FNN (FNN10); a three-layer convolutional neural network (CNN3), a five-layer CNN (CNN5), and a ten-layer CNN (CNN10); a three-layer Long Short-Term Memory (LSTM) RNN model and a three-layer Gated Recurrent Unit (GRU) RNN model. These network architectures offer flexibility for complex data representations. Fig 3 outlines the main features of the FNN, CNN, and RNN networks used in our study.

## 1.5. Model hypertuning

We present hereinafter the methodology used for hypertuning the traditional ML classifiers and DL network models considered in our study.

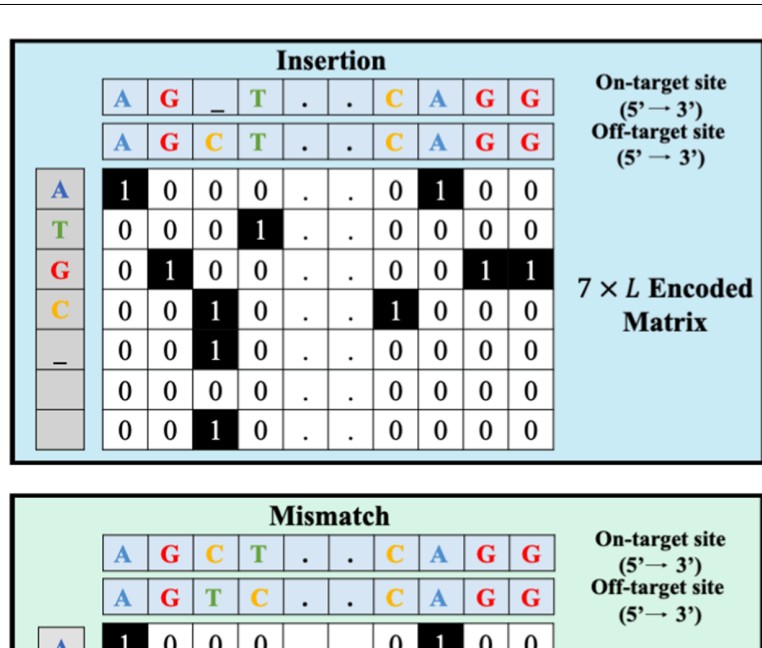

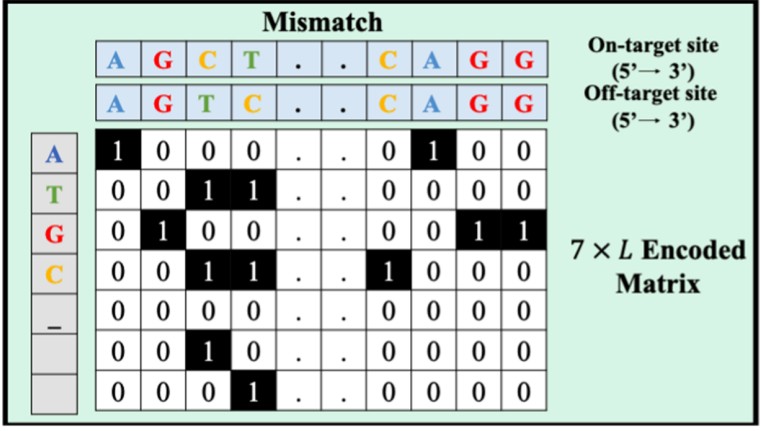

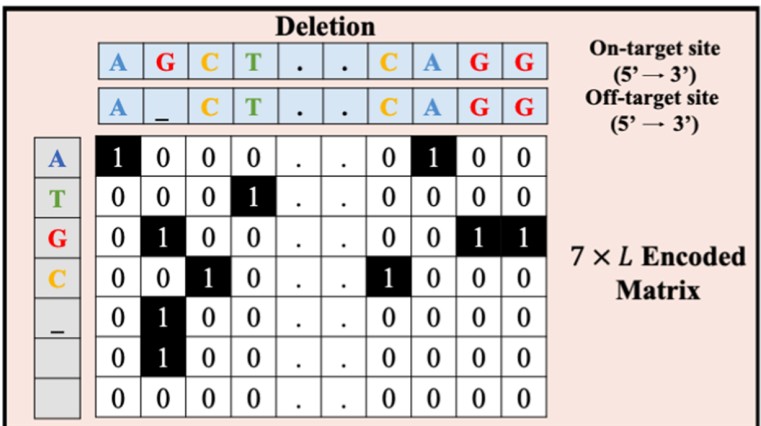

**Fig 2. A schematic view of the encoding of an sgRNA-DNA sequence pair, as employed in the study of Lin et al.** [37]. A seven-bit encoding example is illustrated, where the _ symbol indicates the position of DNA or RNA bulges. Each sgRNA-DNA sequence pair is encoded as a fixed-length matrix with seven rows, comprising a five-bit character channel (A, G, C, T, _) and a two-bit direction channel. The five-bit channel encodes the nucleotides at the on- and off-target sites, while the direction channel identifies the locations of mismatches and indels. $L$ denotes the sequence length ($L$=23 in our study).

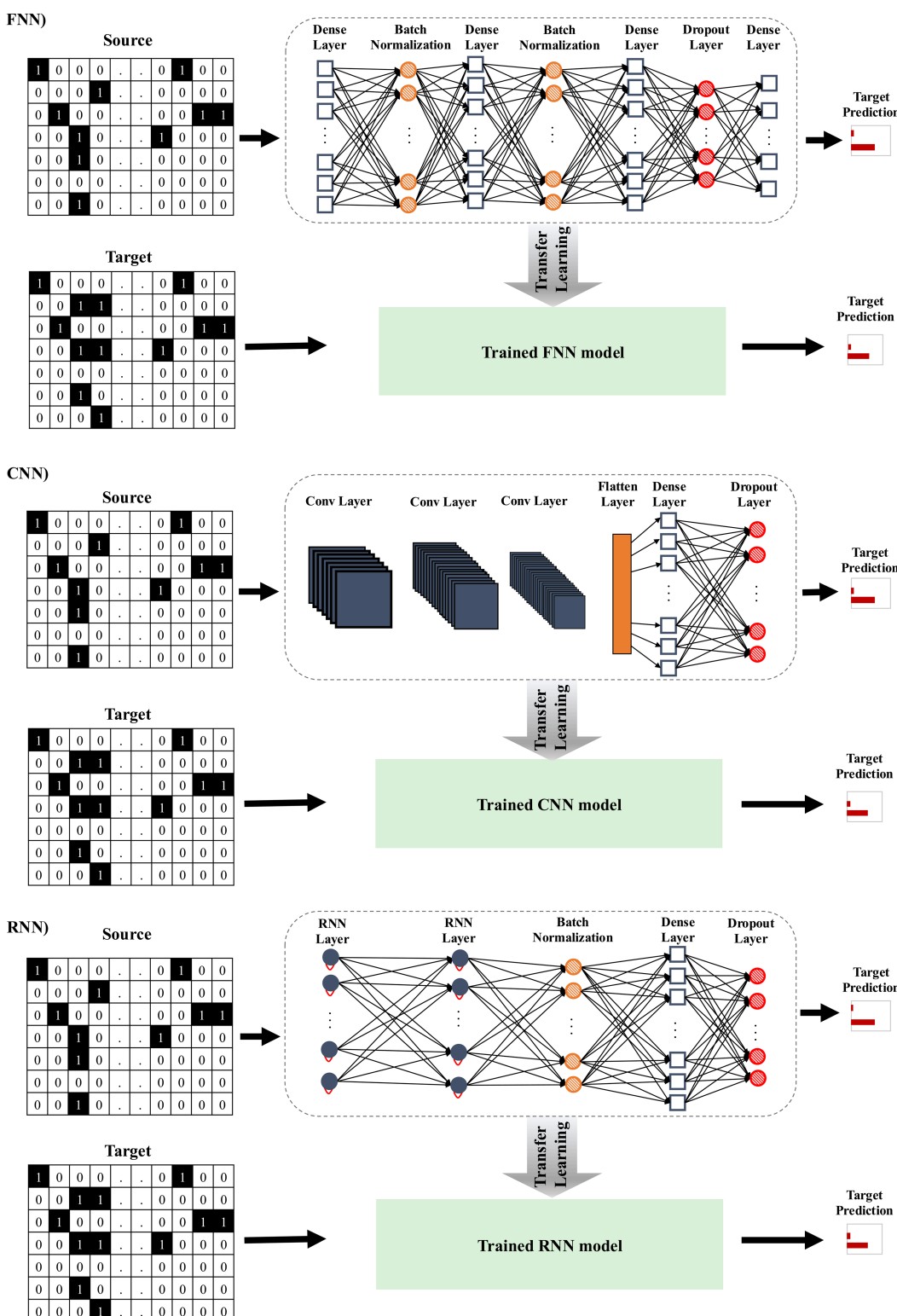

**Fig 3. Representation of transfer learning for FNNs, CNNs, and RNNs.** Minor variations exist between different CNN and FNN architectures used in our experiments, based on the number of layers included. However, the architecture presented is consistent across all RNNs evaluated in our study.

**1.5.1. Classifiers hypertuning.** To determine the optimal parameters for each trained classifier, we employed random search with a 3-fold cross-validation on the training set. The 3-fold cross-validation ensures that there is no data leakage when assessing performance on the test set in our experiments. We utilized the *RandomizedSearchCV* function from the Scikit-Learn library, which implements a random search. Unlike exhaustive grid search, random search explores a subset of hyperparameter values using a fixed number of samples. These values can be specified as lists or sampled from distributions. By doing so, random search efficiently explores a wider range of hyperparameters while minimizing the running time [46]. The hyperparameters used with the CD33, CIRCLE, and SITE datasets are detailed in Tables A, B, and C in S1 Table.

**1.5.2. Deep neural networks hypertuning.** Finding the optimal set of hyperparameters for DL models can require significant computational time and resources [46]. Thus, we decided to use a random search method optimized for DL models offering a good compromise between the computational resources being employed and the optimality of the model's parameters [47]. The Keras Tuner library offers a simple and efficient framework to fine-tune the deep learning models used in our experiments: FNN3, FNN5, FNN10, CNN3, CNN5, CNN10, LSTM, and GRU. With DL models, we applied a 3-fold cross-validation on the training set applying a similar methodology as that used with traditional scikit-learn classifiers. The number of maximum trials was set to 30. This parameter represents the maximum total number of trials during a hyperparameter search. The hyperparameters for the CD33, CIRCLE, and SITE datasets are detailed in Tables D, E, and F in S1 Table.

## 1.6. Neural network overfit monitoring

Both traditional ML classifiers and DL networks require overfit monitoring during their training [46]. In our experiments, we employed two essential callbacks to mitigate overfitting in our deep learning models. First, we used the Reduce Learning Rate on Plateau callback dynamic approach that automatically adjusts the learning rate during training based on the model's performance. If the validation loss reaches a plateau (i.e., stops improving), the learning rate is reduced, allowing the model to converge more effectively. Second, we used the Early Stopping callbacks with a patience of 8 epochs and a minimum *delta* parameter of 0.02 on the validation loss. This callback function monitors the model's performance during training. If the validation loss fails to improve significantly (i.e. it is less than the specified *delta*) for a certain number of consecutive epochs (determined by the patience value), the training is halted early to prevent overfitting. These combined strategies help ensure that our DL models generalize well to unseen data [48].

## 1.7. Transfer learning based on distance evaluation

In this section, we delve into the background and detailed explanation of the proposed approach for similarity-based transfer learning off-target prediction in CRISPR-Cas9. A notable concern in transfer learning is the risk of negative transfer [49], which arises when the source dataset is inappropriately selected. In this case, a model pre-trained on a larger but dissimilar dataset may perform worse than a model trained from scratch with randomly initialized weights. This issue emphasizes the importance of quantifying the similarity between source and target datasets to ensure the success of transfer learning. To address this challenge for off-target CRISPR-Cas9 data, we evaluate the similarity between the two involved datasets (i.e. a potential source and the given target datasets) using three different

metrics: cosine similarity (here, we use its distance form), and Euclidean and Manhattan distances:

$$d_{cosine}(\mathbf{a}, \mathbf{b}) = 1 - \frac{\sum_{i=1}^{K} a_i b_i}{\sqrt{\sum_{i=1}^{K} a_i^2}\sqrt{\sum_{i=1}^{K} b_i^2}}, \tag{1}$$

$$d_{Euclidean}(\mathbf{a}, \mathbf{b}) = \sqrt{\sum_{i=1}^{K}(a_i - b_i)^2}, \tag{2}$$

$$d_{Manhattan}(\mathbf{a}, \mathbf{b}) = \sum_{i=1}^{K} |a_i - b_i|. \tag{3}$$

Here, $\mathbf{a}$ and $\mathbf{b}$ are vectors of length $K = 7L$ (i.e. $L$ is the sequence length, which is equal to 23 in our study) representing the encoded sgRNA-DNA sequence pairs in the source and target datasets, respectively. Each encoded matrix of size $7 \times L$ (see Fig 2) is flattened into a vector of length $7L$ in order to calculate the distance between sgRNA-DNA sequence pairs from the source and target datasets. It is worth noting that the number of rows, i.e. 7, in the matrix corresponds to the number of bits used to encode a given sgRNA-DNA sequence pair, including a five-bit character channel (i.e. A, G, C, T, and _) and a two-bit direction channel (to indicate if they appear in the sgRNA or DNA sequence). The five-bit channel encodes the presence-absence of the four nucleotides and insertions/deletions, whereas the direction channel identifies the location of the mismatched nucleotides or insertions/deletions, if any (see Fig 2 for three examples of such a sequence encoding). We should highlight that 7 is the minimum number of rows one can use to encode the presence-absence of the A, G, C, T, and _ characters in two sequences, including their location information, without any information loss.

It is important to note that cosine, Euclidean, and Manhattan distances compared in our study have different strengths and disadvantages. For example, cosine distance is sensitive to direction in high-dimensional sparse spaces. It is used when the magnitude of the vectors is not important. This is the case of binary data such as our sgRNA-DNA sequence pairs encoded as $7L$ binary vectors. With cosine distance the difference in values is not fully taken into account, but this is not a disadvantage in our settings since this difference can be either 0 or 1 with our sequence encoding. Theoretically, cosine distance should work better in cases when the encoded sgRNA-DNA sequence pairs have more matching nucleotides, than in cases with frequent insertions, deletions, and mismatches, since better matching sequences would lead to more sparse binary spaces. Euclidean distance is certainly the most natural distance choice as it is directly computed from the cartesian coordinates of the points using the Pythagorean theorem. Euclidean distance is sensitive to magnitude but, as specified above, this is not of importance in our binary settings. This distance usually works well with low-dimensional data. Manhattan distance is known for its robustness to outliers. It is less intuitive than Euclidean distance but works well with discrete and binary components as it considers the veritable path that can be taken within values of those components. Thus, in our settings, we could expect that Manhattan distance treats equally well binary encoded matching nucleotides, insertions, deletions, and mismatches.

Clearly, the most intuitive way of computing the distance between the source and the target datasets is the following:

- A. For each element in the source dataset, calculate the minimum distance between it and all elements in the target dataset.
- B. For each element in the target dataset, calculate the minimum distance between it and all elements in the source dataset.
- C. Take an average of all these minimum distances. This average can play the role of the distance between the source and the target data. This distance could then be used to determine the most appropriate source dataset for a given target dataset to carry out transfer learning.

Such an exhaustive approach would work in practice when both the source and target datasets are small ($< 50,000$ elements each). However, real-world CRISPR-Cas 9 datasets often contain hundreds of thousands elements (see Table 1) each of which must be encoded in a numerical vector format beforehand in order to perform machine learning experiments. For example, the execution of the above-mentioned exhaustive approach applied to the CIRCLE (used as source data) and SITE (used as target data) datasets would require several weeks of intensive computation on a modern PC computer. Moreover, in many practical situations the target datasets is so small that deep learning experiments, which usually necessitate a huge amount of data, cannot be performed on it (e.g. see [25,28] for examples of such off-target datasets used in CRISPR-Cas9). Finally, to perform our Monte Carlo simulations to determine the most appropriate distance measure as well as the most suitable ML and DL models in the context of CRISPR-Cas9 off-target transfer learning, we need several hundred real-world datasets of realistic size.

Thus, we decided to perform our Monte Carlo simulations with bootstrap replicates of the considered benchmark target datasets. A bootstrap replicate of a given target dataset per similarity experiment was generated and the average simulation results were then reported. The size of each bootstrapped target dataset was 250, while the number of iterations (i.e. comparisons of each target element with the source elements used to assess the distance between the source and target datasets) was set to 5,000. As we determined experimentally, with this number of iterations (denoted by $n_{itr}$ in Algorithm 1 below), the average distance between the source and target datasets found by the random search converges towards the distance provided by an exhaustive search algorithm. In the large majority of cases, this number of iterations was sufficient to achieve two-digit precision after the decimal point during the distance calculation. This allowed us to obtain reliable results without having to run the computations for several days. Regarding the size of 250 of the bootstrapped targets, it was selected to see how the proposed methodology would work with some high-quality CRISPR-Cas9 datasets of realistically small size. For example, in a recent Nature Communication paper, Ham et al. [50] used the TevSpCas9 dataset with 279 samples as well as the SpCas9 dataset with 303 samples to conduct their transfer learning experiments with a novel machine learning architecture (crisprHAL) meant to improve sgRNA activity prediction.

Furthermore, we made sure that the sample ratios between the majority and minority classes in each bootstrapped target datasets were equivalent to those in the complete target dataset.

Algorithm 1 outlines the key steps of our similarity-based transfer learning approach. The algorithm takes as input $N$ potential source datasets (representing labeled data), denoted as $\mathcal{D}_S := \{D_1, ...., D_N\}$, a given bootstrapped target dataset (representing unlabeled data), denoted as $D_{\mathcal{T}}$, and a distance measure $d$ (cosine, Euclidean, or Manhattan distance, in our case).

The set of the encoded vector representations of the source dataset $D_i$ ($i = 1, ..., N$) is denoted as $\{\mathbf{X^i}\}$, and of the bootstrapped target dataset $D_{\mathcal{T}_b}$ as $\{\mathbf{X^t}\}$, each of them of dimension $7L$.

**Algorithm 1** Similarity-Based Transfer Learning for CRISPR-Cas9 Off-Target Prediction

**Require:** Set $D_S$ of $N$ potential source datasets $\{D_1, ...., D_N\}$ (labeled off-target data)
**Require:** Bootstrapped target dataset $D_{\mathcal{T}_b}$ (unlabeled off-target data)
**Require:** Distance measure $d$ (i.e. cosine, Euclidean, or Manhattan distance)
**Ensure:** Off-target predictions by similarity-based transfer learning

1: $\mathcal{D}_S := \{\mathcal{D}_1, ...., \mathcal{D}_N\}, \mathcal{D}_i = \text{Encode}(D_i) = \{\mathbf{X^i}\}, \quad i = 1, ...N$
2: $D_{\mathcal{T}_b} = \text{Bootstrap}(\text{Encode}(D_{\mathcal{T}})) = \{\mathbf{X^t}\}$

---

**Phase 1 - Similarity Analysis - Selecting Optimal Source Dataset**

3: $Dist_{Opt} \leftarrow \infty$ ▷ Initialize optimal (minimum) distance between source and target
4: $D_{S_{Opt}} = \varnothing$ ▷ Initialize optimal source dataset for transfer learning
5: **for each** $\mathcal{D}_i \in D_N$ **do**
6: **for** $m \leftarrow 1, ..., |D_{\mathcal{T}_b}|$ **do**
7: $dist_{min} \leftarrow \infty$ ▷ Initialize the minimum distance
8: $\mathbf{a}_m = \mathbf{X^t}_{m,:}$ ▷ Extract the $m^{th}$ sample of the target dataset
9: **for** $iteration \leftarrow 1, ..., n_{itr}$ **do**
10: $n = \text{randint}[1, |\mathcal{D}_i|]$ ▷ Randomly sample an index from the source dataset
11: $\mathbf{b}_n = \mathbf{X^i}_{n,:}$ ▷ Extract the $n^{th}$ sample of the $i^{th}$ source dataset
12: $d_{current} \leftarrow d(\mathbf{a}_m, \mathbf{b}_n)$ ▷ Using Eqs. (1)-(3)
13: **if** $d_{current} < dist_{min}$ **then**
14: $dist_{min} \leftarrow d_{current}$
15: **end if**
16: **end for**
17: $\mathbf{d}_m \leftarrow dist_{min}$ ▷ Store the minimum distance for the $m^{th}$ sample in vector $\mathbf{d}$
18: **end for**
19: **if** $\bar{\mathbf{d}} < Dist_{Opt}$ **then** ▷ $\bar{\mathbf{d}}$ is the mean of the minimum distance vector $\mathbf{d}$
20: $Dist_{Opt} \leftarrow \bar{\mathbf{d}}$ ▷ Update the optimal distance between source and target
21: $D_{S_{Opt}} \leftarrow \mathcal{D}_i$ ▷ Update the optimal source dataset
22: **end if**
23: **end for**

---

**Phase 2 - Transfer Learning**

24: $\mathcal{M}_S \leftarrow$ Train model using the selected source dataset $D_{S_{Opt}}$
25: $w_S \leftarrow$ Save the trained model weights
26: $\mathcal{M}_{\mathcal{T}_S} \leftarrow$ Apply transfer learning using target data by loading weights $w_S$
27: Perform off-target predictions using $\mathcal{M}_{\mathcal{T}_S}$

During the first phase of Algorithm 1, refereed to as Similarity Analysis phase, we systematically evaluate the distance between each potential source dataset $\mathcal{D}_i \in D_N$ and the bootstrapped target dataset $D_{\mathcal{T}_b}$ to determine the most suitable source dataset for a given target. Cosine, Euclidean, or Manhattan distance between the following vectors is then computed: (1) $\mathbf{a}_m$ - a vector of length $7L$ representing the $m^{th}$ encoded sgRNA-DNA sequence pair in the bootstrapped target dataset $D_{\mathcal{T}_b}$, where $m \in \{1, ..., |D_{\mathcal{T}_b}|\}$ and (2) $\mathbf{b}_n$ - a randomly selected vector of length $7L$ representing the $n^{th}$ encoded sgRNA-DNA sequence pair in the $i^{th}$ source dataset $\mathcal{D}_i$, where $n \in \{1, ..., |\mathcal{D}_i|\}$. For each dataset $\mathcal{D}_i \in \mathcal{D}_S$, we iterate over every element in $D_{\mathcal{T}_b}$ computing the current distance value, $d_{current}$, between each element in $D_{\mathcal{T}_b}$ (i.e. $\mathbf{a}_m$ vectors) and a random subset of elements in $\mathcal{D}_i$ (i.e. $\mathbf{b}_n$ vectors). The number of elements in this random subset equals $n_{itr}$. Thus, a unique subset of the current source dataset $\mathcal{D}_i$ is generated through random sampling with replacement. If the computed current distance $d_{current}$ is smaller than the previously stored minimum distance $dist_{min}$, we update $dist_{min}$ to $d_{current}$. This process is repeated over $n_{itr}$ iterations, ensuring that $dist_{min}$ consistently represents a close match between the target element and the source dataset $\mathcal{D}_i$. We then construct the vector $\mathbf{d}$ of dimension $|D_{\mathcal{T}_b}|$ that includes the minimum distance values for all samples in $D_{\mathcal{T}_b}$.

The mean of this vector is used to determine the optimal source candidate dataset for transfer learning (i.e. the set that exhibits the highest similarity with a given target).

The second phase of Algorithm 1, referred to as Transfer Learning phase, implements the transfer learning process using the optimal source dataset $D_{S_{Opt}}$ (selected at Phase 1).

## 2. Results and discussion

In this section, we present the results of our simulation study that addresses two core objectives: (i) evaluating the effectiveness of transfer learning in improving off-target predictions in CRISPR-Cas9, and (ii) developing a methodology for pre-assessing the success of transfer learning predictions through a similarity-based analysis of the source and target data. The flowchart of our approach is illustrated in Fig 1.

### 2.1. Similarity analysis

Similarity analysis evaluates the closeness of a given target dataset to a potential source dataset. This analysis is crucial for determining the appropriateness of employing transfer learning for the data at hand. In our study, cosine, Euclidean, and Manhattan distances were used to quantify the degree of similarity between datasets. Each of these metrics has its own strengths and weaknesses. Thus, cosine distance is preferable for high-dimensional and text data, Euclidean distance provides an intuitive measure of similarity for normalized data, whereas Manhattan distance is beneficial for datasets encompassing outliers or non-linear relationships [51].

To conduct our simulation study, we selected the CD33, CIRCLE, and SITE datasets as candidate sources datasets, as they offer a sufficient number of minority class samples (i.e. off-targets), thereby increasing the robustness of our approach. In our simulations, the size of the subset of the source dataset compared to the given target dataset was set to 5,000 (i.e. $n_{itr}$ = 5,000 in Algorithm 1), whereas the size of the bootstrapped target datasets was set to 250 (the class imbalance ratio in the bootstrapped datasets was equivalent to that of the corresponding complete dataset; see Table 2). We observed that the distance estimations usually converged when the number of iterations, i.e. $n_{itr}$, was between 4000 and 5000.

Table 3 reports the average estimated similarities between the three source datasets (CD33, CIRCLE, and SITE) and the seven bootstrapped target datasets (CD33_BS, CIRCLE_BS, SITE_BS, Tasi_GUIDE_BS, Listgarten_GUIDE_BS, Kleinstiver_GUIDE_BS, and Hmg_BS) calculated using cosine, Euclidean, and Manhattan distances. Each similarity estimate appearing in Table 3 was computed as $1 - NormalizedAverageDistance$ between the selected source and bootstrapped target dataset using cosine, Euclidean, or Manhattan distance. The exact procedure used in our experiments to compute the similarity values is as follows:

**Table 2**. Minority and majority class distribution, and class imbalance ratio for bootstrapped target datasets, with sample size of 250, used in our experiments.

| Dataset | Minority Class Samples | Majority Class Samples | Class Imbalance Ratio |
|---|---|---|---|
| CD33_BS | 117 | 133 | 0.879 |
| CIRCLE_BS | 3 | 247 | 0.012 |
| SITE_BS | 4 | 246 | 0.016 |
| Tasi_GUIDE_BS | 2 | 248 | 0.008 |
| Listgarten_GUIDE_BS | 2 | 248 | 0.008 |
| Kleinstiver_GUIDE_BS | 2 | 248 | 0.008 |
| Hmg_BS | 3 | 247 | 0.012 |

**Table 3**. **Average Estimated Similarities (1 - Normalized Average Distances) between the three source datasets (CD33, CIRCLE, and SITE) and the seven bootstrapped target datasets (CD33_BS, CIRCLE_BS, SITE_BS, Tasi_GUIDE_BS, Listgarten_GUIDE_BS, Kleinstiver_GUIDE_BS, and Hmg_BS) calculated using cosine, Euclidean, and Manhattan distances.** Similarity values corresponding to the most suitable source-target dataset pairs are highlighted in bold.

| Target data | Metric | CD33 | CIRCLE | SITE |
|---|---|---|---|---|
| CD33_BS | Cosine | **0.9585** | 0.5669 | 0.5484 |
| | Euclidean | **0.7843** | 0.0821 | 0.0520 |
| | Manhattan | **0.8933** | 0.1589 | 0.1095 |
| CIRCLE_BS | Cosine | 0.5443 | **0.8650** | 0.6299 |
| | Euclidean | 0.0680 | **0.4450** | 0.0845 |
| | Manhattan | 0.1295 | **0.6809** | 0.1389 |
| SITE_BS | Cosine | 0.5352 | 0.6021 | **0.8841** |
| | Euclidean | 0.0408 | 0.0589 | **0.4550** |
| | Manhattan | 0.0884 | 0.10280 | **0.7052** |
| Tasi_GUIDE_BS | Cosine | 0.5448 | **0.8269** | 0.6266 |
| | Euclidean | 0.0549 | **0.3586** | 0.0776 |
| | Manhattan | 0.1105 | **0.5755** | 0.1283 |
| Listgarten_GUIDE_BS | Cosine | 0.5605 | **0.5701** | 0.5583 |
| | Euclidean | **0.0726** | 0.0202 | 0.0024 |
| | Manhattan | **0.1563** | 0.0502 | 0.0080 |
| Kleinstiver_GUIDE_BS | Cosine | 0.5265 | **0.5672** | **0.5676** |
| | Euclidean | **0.0418** | 0.0184 | 0.0007 |
| | Manhattan | **0.0827** | 0.0329 | 0.0220 |
| Hmg_BS | Cosine | 0.5317 | 0.5598 | **0.5728** |
| | Euclidean | **0.0695** | 0.0442 | 0.0363 |
| | Manhattan | **0.1478** | 0.0725 | 0.0616 |

1. A bootstrapped target dataset of size 250 was generated for each of the 7 (complete) benchmark off-target datasets considered in our work. The class imbalance ratio of the corresponding complete benchmark dataset was preserved in bootstrapped data (e.g. see Table 2);

2. A $7 \times 7$ distance matrix, **Dist**, containing pairwise distances between 7 complete and 7 bootstrapped datasets was computed using Algorithm 1;

3. Steps 1 and 2 above were repeated 5 times to create 5 replicates of the distance matrix **Dist**;

4. The average $7 \times 7$ distance matrix **Dist_av** was computed using these replicates;

5. The average similarity matrix **S** was computed from this average distance matrix using the Min-Max normalization:

$$s(i,j) = 1 - \frac{Dist\_av(i,j) - min(Dist\_av)}{max(Dist\_av) - min(Dist\_av)},$$

where $1 \leq i,j \leq 7$, and $max(Dist\_av)$ and $min(Dist\_av)$ are, respectively, the minimum and maximum values of the distance matrix **Dist_av**. Obviously, higher similarity values are associated with lower distances.

In addition, Fig 4 presents the corresponding bar plot diagrams for each of the three considered distance measures.

The results presented in Table 3 and Fig 4 demonstrate that the cosine metric provides the highest overall similarity values between source and target datasets, compared to Manhattan and Euclidean distances, whereas Euclidean distance corresponds to the lowest similarities. However, Manhattan and Euclidean metrics provide the largest differences between the similarities corresponding to the recommended and non-recommended source datasets. This means that Euclidean and Manhattan distances, being sensitive to absolute differences, better

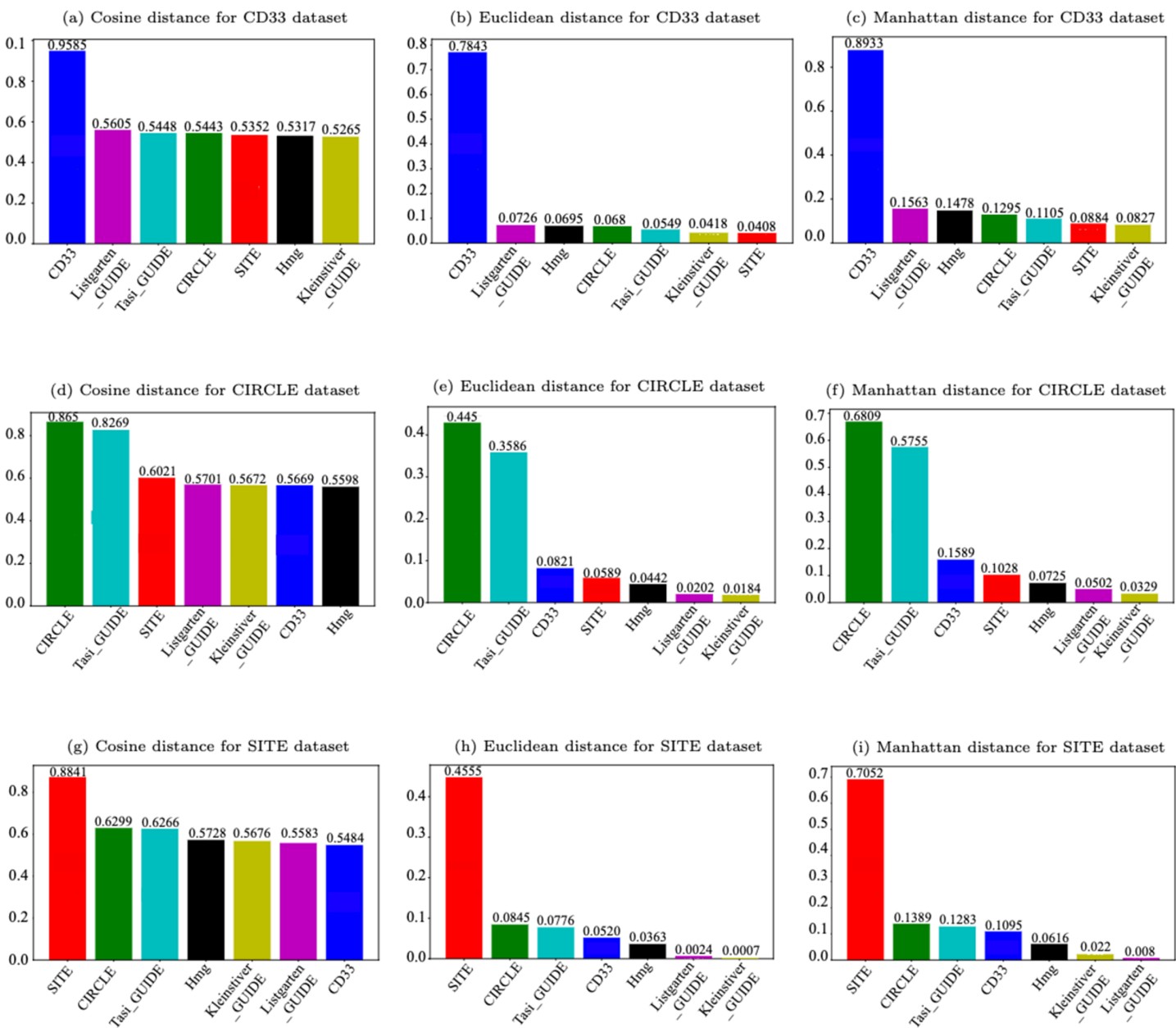

**Fig 4. Bar plot representation of Average Estimated Similarities (1 - Normalized Average Distances).** Similarities between the three source datasets (CD33, CIR-CLE, and SITE) and the seven bootstrapped target datasets (CD33_BS, CIRCLE_BS, SITE_BS, Tasi_GUIDE_BS, Listgarten_GUIDE_BS, Kleinstiver_GUIDE_BS, and Hmg_BS) were assessed using the cosine, Euclidean, and Manhattan distances.

highlight stark dissimilarity between datasets. However, our findings suggest that overall magnitude is less important for transferability than feature direction, which is captured by cosine distance (see Sect 2.3). Such a result should be related to the binary nature of the encoded sgRNA-DNA sequence pairs since in our settings even the vectors with different locations of matches, mismatches, and indels are sparse enough to have at least 50% of matching 0 values, thus leading to the lowest cosine similarity values that are slightly higher than 0.5. In the case of Euclidean and Manhattan distances, the lowest normalized distance values can be close to

1, leading to the corresponding similarity values that are slightly higher than 0. Obviously, the maximum similarity value is limited by 1 (in case of a perfect sequence match) for all three metrics considered.

As expected, the most appropriate source datasets for the bootstrapped target datasets CD33_BS, CIRCLE_BS, and SITE_BS were their complete source counterparts CD33, CIRCLE, and SITE, respectively. The corresponding cosine similarity values for these source-target pairs were 0.9585, 0.8650, and 0.8841, respectively.

Interestingly, for the other target datasets, i.e. Tasi_GUIDE_BS, Listgarten_GUIDE_BS, Kleinstiver_GUIDE_BS, and Hmg_BS, the choice of the most suitable source dataset depends on the selected distance/similarity measure. For example, for Tasi_GUIDE_BS, the CIRCLE dataset stands out as the most suitable source, achieving the highest similarity across all three distance metrics (cosine, Euclidean, and Manhattan). However, for Listgarten_GUIDE_BS, the CIRCLE dataset is the most suitable source according to cosine similarity, but both Manhattan and Euclidean metrics indicate CD33 as the most suitable source dataset for it. In the case of Kleinstiver_GUIDE_BS, the CIRCLE and SITE datasets provide a slightly better performance compared to CD33 according to cosine similarity. However, according to both Euclidean and Manhattan metrics, CD33 shows the highest similarity with this target dataset. For Hmg_BS, the obtained results reveal that the SITE dataset demonstrates the highest similarity with it using cosine similarity, whereas the CD33 dataset is designated as the most suitable source for it according to Euclidean and Manhattan metrics. Clearly, the choice of the most appropriate source dataset depends on the specific distance/similarity measure being employed as the results provided by Euclidean and Manhattan metrics are usually well aligned, but don't always correspond to those yielded by the cosine metric.

## 2.2. Evaluation metrics

The performance of the proposed model was assessed using several standard evaluation metrics, as detailed below:

- **AUC_ROC (Area Under the Receiver Operating Characteristic Curve):** This metric evaluates the model's ability to distinguish between positive and negative classes. It represents the probability that the classifier will assign a higher score to a randomly chosen positive instance than to a randomly chosen negative instance.
- **Precision** is defined as the proportion of true positive predictions among all predicted positives:

$$\text{Precision} = \frac{TP}{TP + FP}, \tag{4}$$

where $TP$ denotes the number of true positives and $FP$ the number of false positives.
- **Recall**, which is also known as sensitivity or true positive rate, assesses the proportion of correctly identified positive samples:

$$\text{Recall} = \frac{TP}{TP + FN}, \tag{5}$$

where $FN$ is the number of false negatives.
- **F1-score** is the harmonic mean of precision and recall, offering a balance between them:

$$\text{F1-score} = 2 \times \frac{\text{Precision} \times \text{Recall}}{\text{Precision} + \text{Recall}}. \tag{6}$$

- **Brier score** is defined as the mean squared difference between the predicted probability and its binary outcome:

$$\text{Brier score} = \frac{1}{N}\sum_{i=1}^{N}(p_i - o_i)^2, \tag{7}$$

where $p_i$ is the predicted probability of sample $i$, $o_i \in \{0, 1\}$ is the observed outcome of $i$, and $N$ is the total number of samples. Lower values of the Brier score correspond to better calibrated probabilistic predictions.
- **Accuracy** is the proportion of correctly classified samples among their total number:

$$\text{Accuracy} = \frac{TP + TN}{TP + TN + FP + FN}, \tag{8}$$

where $TN$ is the number of true negatives.

## 2.3. Assessing the impact of similarity analysis in transfer learning

In this section, we evaluate the impact of similarity analysis in transfer learning with CRISPR-Cas9 off-target data. Thus, we assess the reliability of the similarity scores reported in Table 3 for ML and DL-based off-target predictions. This evaluation is structured around three distinct scenarios, where machine learning models are trained on one of the three source datasets (CD33, CIRCLE, and SITE) and applied, via transfer learning, to different variants of seven bootstrapped datasets constructed as outlined in Table 2. For each scenario, we report the average results for 10 DL models: FNN models with 3, 5, and 10 layers; CNN models with 3, 5, and 10 layers; LSTM and GRU models with 4 layers; MLP models with 1 and 2 layers; as well as for traditional RF and LR classifiers (see Supplementary Information for further details on the models considered).

In the first scenario, the CD33 dataset served as the source dataset for transfer learning. The Receiver Operating Characteristic (ROC) curves (see Fig 5A) and Precision-Recall (PR) curves (see Fig A(A) in S1 Fig) are presented for various models trained on the CD33 dataset and tested on its bootstrapped counterpart, CD33_BS. To help evaluate the models performance, the AUC-ROC values are displayed in descending order within these figures. Furthermore, Fig 6 and Fig B in S1 Fig present the ROC and PR curves, respectively, for all considered ML and DL models using the CD33 dataset as source and six other bootstrapped datasets as targets. Additionally, Table 4 reports the values of the six selected evaluation metrics, including AUC ROC, Precision, Recall, F1-score, Brier score, and Accuracy, obtained using the CD33 dataset as source for all bootstrapped targets. Target datasets exhibiting the highest similarity to the CD33 dataset are marked with an asterisk (based on the similarity scores reported in Table 3). When the CD33_BS dataset served as the transfer learning target, GRU and MLP1 achieved a superior AUC-ROC performance compared to other models, with MLP1 providing the highest AUC-ROC score of 0.9863, closely followed by GRU at 0.9839. Both GRU and MLP1 consistently outperformed the other competing models across all evaluation metrics. Among the bootstrapped datasets, Listgarten_GUIDE_BS, Kleinstiver_GUIDE_BS, and Hmg_BS showed the highest similarity with CD33 according to Euclidean and Manhattan metrics (see Table 3). When Listgarten_GUIDE_BS was used as target, MLP1 achieved the highest AUC ROC (0.9629) and Precision (0.6828) results. When Kleinstiver_GUIDE_BS was used as target, MLP1 and MLP2 outperformed all other models across all metrics. Moreover, MLP2 consistently provided the best results across all metrics for the Hmg_BS target dataset, with the highest AUC-ROC, precision, and F1-score values, and the lowest Brier score.

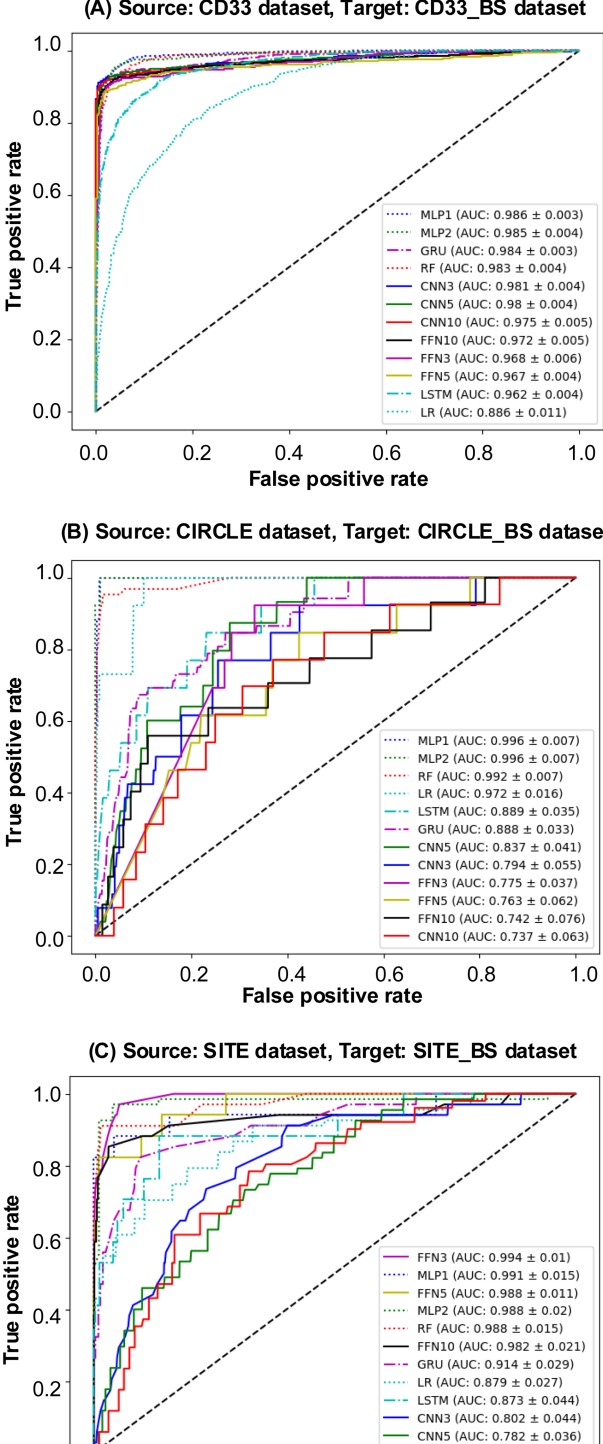

**Fig 5. ROC curves for model evaluation.** ROC curves for models trained on: (A) CD33 dataset, (B) CIRCLE dataset, and (C) SITE dataset, used as sources, and evaluated on their respective bootstrapped targets. The AUC ROC values for each model are displayed in descending order within each figure.

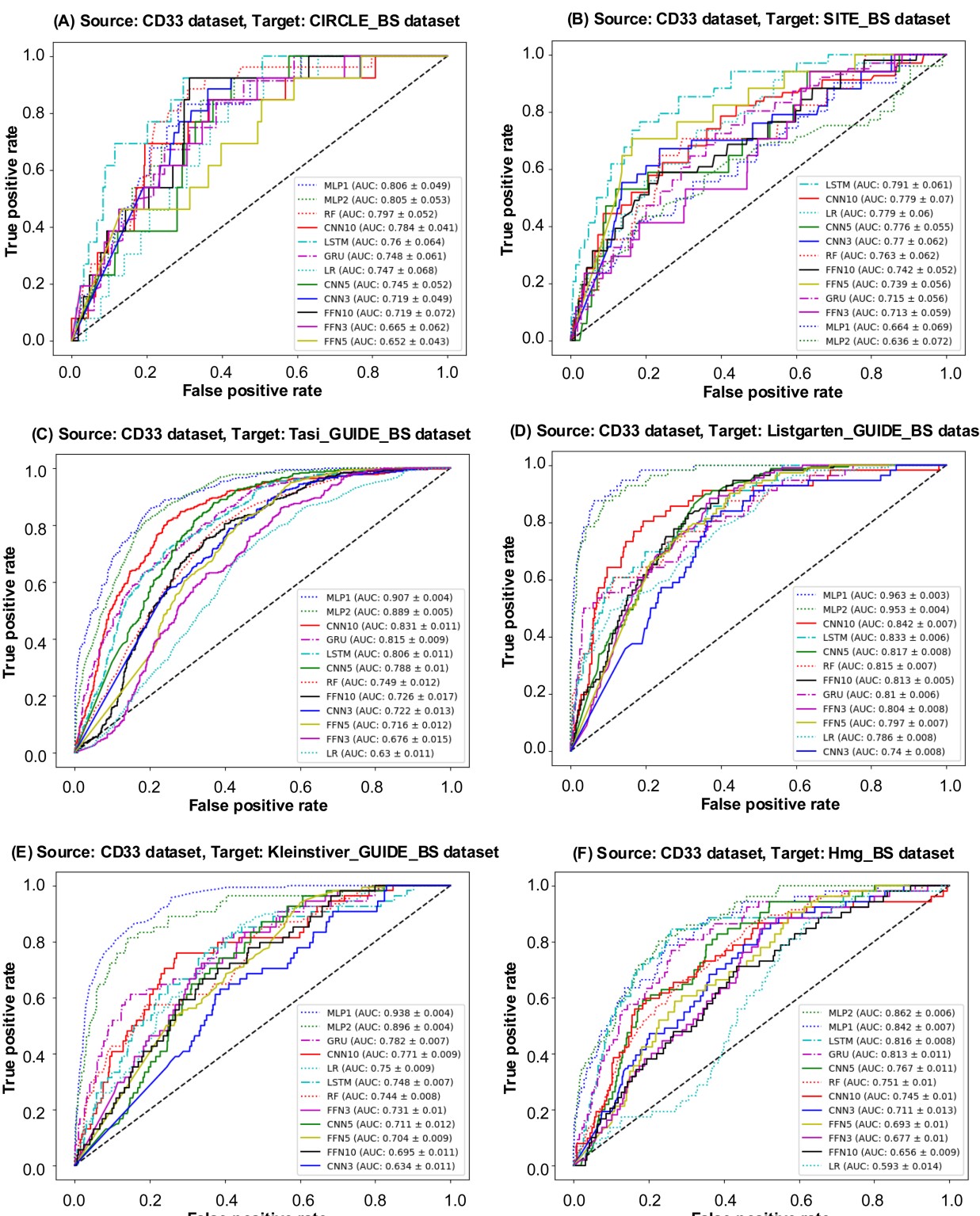

**Fig 6. ROC curves for model evaluation.** ROC curves for models trained on the CD33 dataset, used as source, and six bootstrapped datasets: CIRCLE_BS, SITE_BS, Tasi_GUIDE_BS, Listgarten_GUIDE_BS, Kleinstiver_GUIDE_BS, Hmg_BS, used as targets. The AUC ROC values for each model are displayed in descending order within the figure.

**Table 4. Performance metrics for each considered classification model obtained using the CD33 dataset for training (i.e. as source).** Target datasets exhibiting the highest similarity to the CD33 dataset are marked with an asterisk. The results of the top-performing models are highlighted in bold.

| Target | Metric | FNN3 | FNN5 | FNN10 | CNN3 | CNN5 | CNN10 | LSTM | GRU | MLP1 | MLP2 | RF | LR |
|---|---|---|---|---|---|---|---|---|---|---|---|---|---|
| CD33_BS* | AUC ROC | 0.9680 | 0.9671 | 0.9720 | 0.9814 | 0.9799 | 0.9754 | 0.9618 | **0.9839** | **0.9863** | 0.9851 | 0.9829 | 0.8860 |
| | Precision | 0.9740 | 0.9745 | 0.9785 | 0.9859 | 0.9842 | 0.9823 | 0.9608 | **0.9867** | **0.9832** | 0.9803 | 0.9822 | 0.8621 |
| | Recall | 0.8782 | 0.8889 | 0.9145 | 0.8931 | 0.8814 | 0.9060 | 0.9299 | **0.9402** | **0.9583** | 0.9487 | 0.9701 | 0.8120 |
| | F1 Score | 0.9278 | 0.9265 | 0.9369 | 0.9425 | 0.9348 | 0.9464 | 0.8838 | **0.9392** | **0.9522** | 0.9541 | 0.9390 | 0.7925 |
| | Brier Score | 0.0570 | 0.0578 | 0.0493 | 0.0481 | 0.0499 | 0.0440 | 0.0874 | **0.0464** | **0.0361** | 0.0371 | 0.0550 | 0.1489 |
| | Accuracy | 0.9360 | 0.9340 | 0.9423 | 0.9490 | 0.9425 | 0.9520 | 0.8856 | **0.9430** | **0.9550** | 0.9573 | 0.9410 | 0.8010 |
| CIRCLE_BS | AUC ROC | 0.6650 | 0.6518 | 0.7189 | 0.7193 | 0.7447 | 0.7840 | 0.7599 | 0.7477 | 0.8056 | 0.8051 | 0.7974 | 0.7467 |
| | Precision | 0.0576 | 0.0445 | 0.0671 | 0.0533 | 0.0667 | 0.1225 | 0.1052 | 0.1040 | 0.1342 | 0.1165 | 0.0951 | 0.0799 |
| | Recall | 0.5384 | 0.4615 | 0.5385 | 0.9231 | 0.8462 | 0.8462 | 0.8846 | 0.4403 | 0.0000 | 0.0170 | 0.3846 | 0.5637 |
| | F1 Score | 0.1356 | 0.0752 | 0.1057 | 0.0721 | 0.0965 | 0.0799 | 0.1365 | 0.12345 | 0.0000 | 0.0282 | 0.1266 | 0.0947 |
| | Brier Score | 0.1703 | 0.2736 | 0.2115 | 0.5926 | 0.3815 | 0.4633 | 0.2392 | 0.1378 | 0.0275 | 0.0282 | 0.1069 | 0.1754 |
| | Accuracy | 0.8185 | 0.7050 | 0.7630 | 0.3820 | 0.5880 | 0.4930 | 0.7070 | 0.8390 | 0.9722 | 0.9705 | 0.8574 | 0.7231 |
| SITE_BS | AUC ROC | 0.6759 | 0.7158 | 0.7256 | 0.7221 | 0.7880 | 0.8313 | 0.8062 | 0.8151 | 0.9067 | 0.8887 | 0.7487 | 0.6295 |
| | Precision | 0.4387 | 0.4925 | 0.4969 | 0.5187 | 0.5899 | 0.6840 | 0.6508 | 0.6683 | 0.8363 | 0.7926 | 0.5625 | 0.4138 |
| | Recall | 0.5156 | 0.6884 | 0.6884 | 0.9632 | 0.9292 | 0.9575 | 0.7652 | 0.6686 | 0.1218 | 0.1132 | 0.5542 | 0.7651 |
| | F1 Score | 0.5098 | 0.5810 | 0.6152 | 0.6066 | 0.6735 | 0.6377 | 0.6526 | 0.6413 | 0.2166 | 0.2022 | 0.5602 | 0.5670 |
| | Brier Score | 0.3338 | 0.3282 | 0.2767 | 0.4232 | 0.2974 | 0.3464 | 0.2415 | 0.2190 | 0.3038 | 0.3083 | 0.2081 | 0.2543 |
| | Accuracy | 0.6500 | 0.6495 | 0.6960 | 0.5590 | 0.6820 | 0.6160 | 0.7120 | 0.7360 | 0.6890 | 0.6843 | 0.6920 | 0.5870 |
| Tasi_GUIDE_BS | AUC ROC | 0.7131 | 0.7387 | 0.7419 | 0.7705 | 0.7762 | 0.7793 | 0.7905 | 0.7152 | 0.6638 | 0.6360 | 0.7632 | 0.7787 |
| | Precision | 0.1156 | 0.0922 | 0.1944 | 0.1127 | 0.1551 | 0.1667 | 0.2499 | 0.1122 | 0.1443 | 0.1227 | 0.1739 | 0.1693 |
| | Recall | 0.2941 | 0.7647 | 0.5686 | 0.7005 | 0.5882 | 0.7487 | 0.7941 | 0.3835 | 0.0000 | 0.0000 | 0.3529 | 0.7647 |
| | F1 Score | 0.1136 | 0.1425 | 0.1400 | 0.0987 | 0.1299 | 0.1163 | 0.1730 | 0.1444 | 0.0000 | 0.0000 | 0.1277 | 0.1197 |
| | Brier Score | 0.1461 | 0.2801 | 0.2045 | 0.4021 | 0.2470 | 0.3335 | 0.2125 | 0.1349 | 0.0336 | 0.0335 | 0.1459 | 0.2321 |
| | Accuracy | 0.8440 | 0.6870 | 0.7633 | 0.5700 | 0.7320 | 0.6162 | 0.7425 | 0.8457 | 0.9663 | 0.9663 | 0.8360 | 0.6175 |
| Listgarten_GUIDE_BS* | AUC ROC | 0.8037 | 0.7974 | 0.8134 | 0.7396 | 0.8169 | 0.8424 | 0.8327 | 0.8098 | **0.9629** | 0.9530 | 0.8148 | 0.7860 |
| | Precision | 0.1416 | 0.1443 | 0.1741 | 0.1119 | 0.1616 | 0.2399 | 0.2560 | 0.3021 | **0.6828** | 0.6027 | 0.2962 | 0.2431 |
| | Recall | 0.5893 | 0.7679 | 0.7143 | 0.9286 | 0.9107 | 0.9107 | 0.6964 | 0.5714 | **0.1429** | 0.1071 | 0.6071 | 0.7857 |
| | F1 Score | 0.2662 | 0.2266 | 0.2459 | 0.1578 | 0.2214 | 0.1882 | 0.2400 | 0.2832 | **0.2500** | 0.1935 | 0.2798 | 0.1832 |
| | Brier Score | 0.1739 | 0.2736 | 0.2199 | 0.5259 | 0.3346 | 0.3846 | 0.2011 | 0.1386 | **0.0450** | 0.0471 | 0.1501 | 0.2311 |
| | Accuracy | 0.8175 | 0.7060 | 0.7540 | 0.4450 | 0.6410 | 0.5600 | 0.7530 | 0.8380 | **0.9520** | 0.9500 | 0.8250 | 0.6075 |
| Kleinstiver_GUIDE_BS* | AUC ROC | 0.7305 | 0.7036 | 0.6947 | 0.6343 | 0.7106 | 0.7713 | 0.7485 | 0.7821 | **0.9379** | 0.8956 | 0.7442 | 0.7502 |
| | Precision | 0.1083 | 0.0967 | 0.1101 | 0.0768 | 0.0917 | 0.1636 | 0.1508 | 0.2459 | **0.5460** | 0.4574 | 0.1871 | 0.1323 |
| | Recall | 0.7222 | 0.7037 | 0.6296 | 0.9074 | 0.8519 | 0.8889 | 0.7778 | 0.6667 | **0.1111** | 0.1481 | 0.6111 | 0.9259 |
| | F1 Score | 0.1862 | 0.1535 | 0.1704 | 0.1266 | 0.1620 | 0.1387 | 0.1888 | 0.1875 | **0.1946** | 0.2500 | 0.1724 | 0.1362 |
| | Brier Score | 0.3251 | 0.3998 | 0.3000 | 0.6531 | 0.4484 | 0.5496 | 0.3101 | 0.2738 | **0.0479** | 0.0460 | 0.1848 | 0.3372 |
| | Accuracy | 0.6590 | 0.5807 | 0.6690 | 0.3241 | 0.5241 | 0.4041 | 0.6390 | 0.6880 | **0.9503** | 0.9521 | 0.6830 | 0.3660 |
| Hmg_BS* | AUC ROC | 0.6770 | 0.6935 | 0.6559 | 0.7106 | 0.7670 | 0.7454 | 0.8157 | 0.8130 | 0.8418 | **0.8620** | 0.7511 | 0.5934 |
| | Precision | 0.0837 | 0.0885 | 0.0833 | 0.1009 | 0.1205 | 0.1455 | 0.1796 | 0.2009 | 0.3617 | **0.3832** | 0.1216 | 0.0705 |
| | Recall | 0.3419 | 0.5962 | 0.3846 | 0.7885 | 0.6538 | 0.7885 | 0.8503 | 0.5748 | 0.1923 | **0.2885** | 0.4423 | 0.6154 |
| | F1 Score | 0.1461 | 0.1722 | 0.1541 | 0.1488 | 0.1771 | 0.1643 | 0.2536 | 0.2706 | 0.2778 | **0.3721** | 0.2125 | 0.1210 |
| | Brier Score | 0.1926 | 0.2770 | 0.1960 | 0.4351 | 0.2864 | 0.3756 | 0.2021 | 0.1397 | 0.0476 | **0.0469** | 0.1752 | 0.2649 |
| | Accuracy | 0.7927 | 0.7020 | 0.7801 | 0.5310 | 0.6833 | 0.5830 | 0.7405 | 0.8395 | 0.9480 | **0.9493** | 0.8295 | 0.5350 |

In the second scenario, the CIRCLE dataset was used as the source dataset in our transfer learning experiments. The corresponding ROC curves (see Fig 5B) and PR curves (see Fig A(B) in S1 Fig) are presented for the considered ML and DL models trained on the entire CIRCLE dataset and evaluated on the CIRCLE_BS dataset. Additionally, Fig 7 and Fig C in S1 Fig show the ROC and PR curves for all considered models obtained using the CIRCLE dataset as source and the six other bootstrapped datasets as targets. The detailed quantitative results are reported in Table 5. When the CIRCLE_BS dataset was used as target, MLP2 performed exceptionally well with an AUC ROC score of 0.9959, a precision of 0.9564, a recall of 0.9231, an F1-score of 0.9600, a Brier score of 0.0021, and an accuracy of 0.9980. When the Tasi_GUIDE_BS dataset was used as target, MLP1 provided the best overal performance across all metrics. Similarly, for the Listgarten_GUIDE_BS target dataset, MLP2 showcased a robust performance, maintained consistency across all evaluation metrics. When the Kleinstiver_BS dataset was used as target, the best overall results were achieved once again using the MLP1 and MLP2 models.

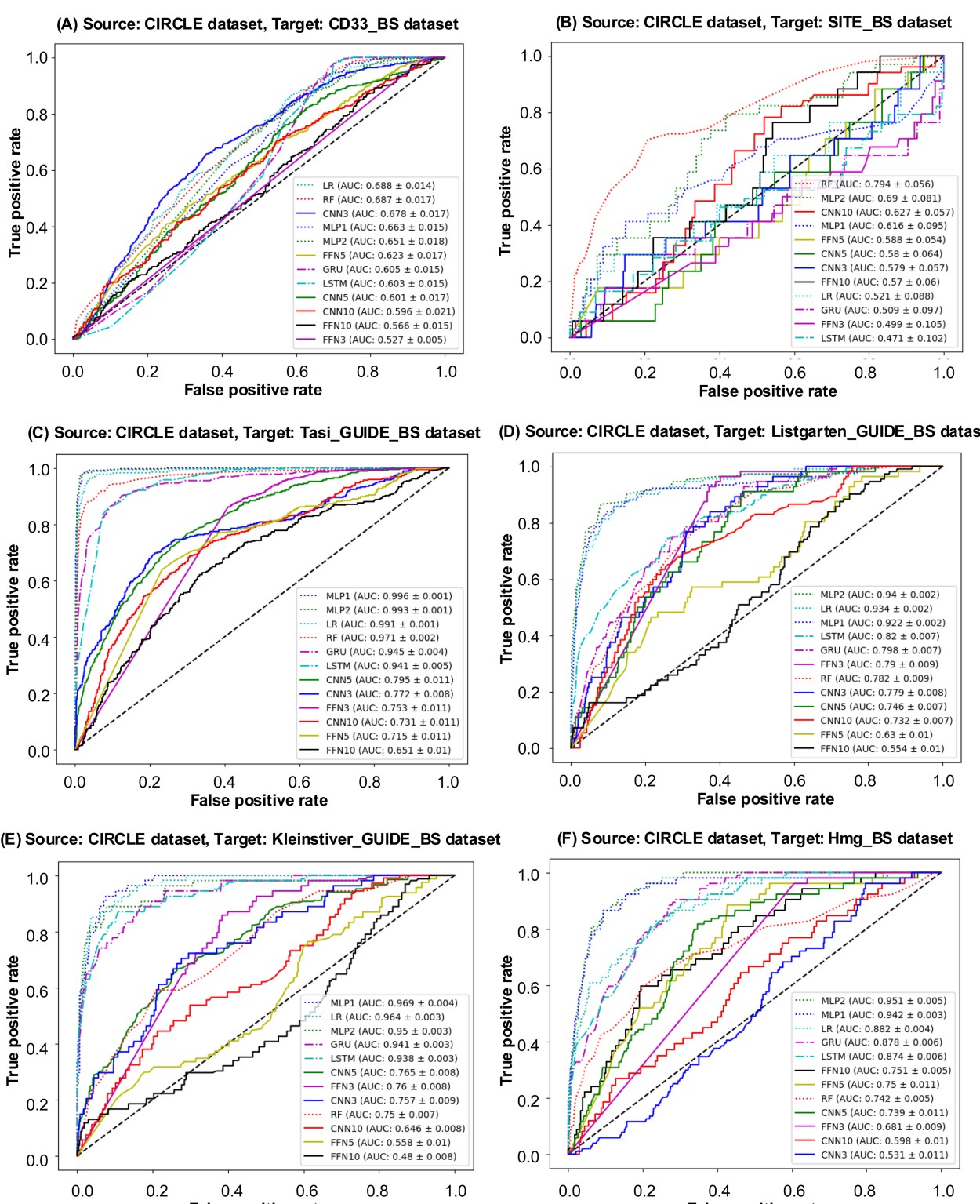

**Fig 7. ROC curves for model evaluation.** ROC curves for models trained on the CIRCLE dataset, used as source, and six bootstrapped datasets: CD33_BS, SITE_BS, Tasi_GUIDE_BS, Listgarten_GUIDE_BS, Kleinstiver_GUIDE_BS, Hmg_BS, used as targets. The AUC ROC values for each model are displayed in descending order within the figure.

**Table 5. Performance metrics for each considered classification model obtained using the CIRCLE dataset for training (i.e. as source).** Target datasets exhibiting the highest similarity to the CIRCLE dataset are marked with an asterisk. The results of the top-performing models are highlighted in bold.

| Target | Metric | FNN3 | FNN5 | FNN10 | CNN3 | CNN5 | CNN10 | LSTM | GRU | MLP1 | MLP2 | RF | LR |
|---|---|---|---|---|---|---|---|---|---|---|---|---|---|
| CD33_BS | AUC ROC | 0.5269 | 0.6227 | 0.5660 | 0.6781 | 0.6014 | 0.5955 | 0.6026 | 0.6053 | 0.6632 | 0.6515 | 0.6872 | 0.6884 |
| | Precision | 0.4818 | 0.5690 | 0.5164 | 0.6055 | 0.5245 | 0.5399 | 0.5159 | 0.5113 | 0.5729 | 0.5673 | 0.6190 | 0.6348 |
| | Recall | 0.9952 | 0.9797 | 1.0000 | 0.9957 | 0.9893 | 0.0085 | 1.0000 | 1.0000 | 1.0000 | 0.9786 | 0.0000 | 0.9979 |
| | F1 Score | 0.6477 | 0.6501 | 0.6407 | 0.6430 | 0.6391 | 0.0167 | 0.6376 | 0.6376 | 0.6624 | 0.6735 | 0.0000 | 0.6518 |
| | Brier Score | 0.3197 | 0.4865 | 0.4543 | 0.5132 | 0.5118 | 0.4009 | 0.5320 | 0.5320 | 0.4676 | 0.4266 | 0.4287 | 0.4843 |
| | Accuracy | 0.4932 | 0.5065 | 0.4750 | 0.4825 | 0.4770 | 0.5290 | 0.4680 | 0.4680 | 0.5230 | 0.5560 | 0.5320 | 0.5010 |
| **CIRCLE_BS\*** | AUC ROC | 0.7754 | 0.7625 | 0.7419 | 0.7939 | 0.8369 | 0.7368 | 0.8888 | 0.8880 | 0.9959 | **0.9956** | 0.9924 | 0.9716 |
| | Precision | 0.0604 | 0.0692 | 0.1300 | 0.1950 | 0.1332 | 0.0762 | 0.2730 | 0.2354 | 0.9385 | **0.9564** | 0.9307 | 0.6556 |
| | Recall | 0.6923 | 0.8462 | 0.8523 | 0.8846 | 1.0000 | 0.1488 | 1.0000 | 1.0000 | 0.8846 | **0.9231** | 0.7551 | 0.4231 |
| | F1 Score | 0.1205 | 0.0742 | 0.0656 | 0.0838 | 0.0778 | 0.0868 | 0.0632 | 0.0633 | 0.9028 | **0.9600** | 0.8503 | 0.5703 |
| | Brier Score | 0.1723 | 0.5441 | 0.5808 | 0.4984 | 0.5994 | 0.0819 | 0.7650 | 0.7660 | 0.0041 | **0.0021** | 0.0058 | 0.0125 |
| | Accuracy | 0.7368 | 0.4510 | 0.3760 | 0.4965 | 0.3898 | 0.9198 | 0.2290 | 0.2298 | 0.9950 | **0.9980** | 0.9934 | 0.9835 |
| SITE_BS | AUC ROC | 0.4990 | 0.5876 | 0.5702 | 0.5786 | 0.5796 | 0.6268 | 0.4711 | 0.5090 | 0.6160 | 0.6902 | 0.7940 | 0.5208 |
| | Precision | 0.0403 | 0.0463 | 0.0462 | 0.0536 | 0.0570 | 0.0530 | 0.0631 | 0.0548 | 0.1544 | 0.1785 | 0.2569 | 0.1148 |
| | Recall | 0.2941 | 0.7059 | 1.0000 | 1.0000 | 0.7647 | 0.0392 | 0.1934 | 0.1765 | 0.0588 | 0.1176 | 0.0701 | 0.1765 |
| | F1 Score | 0.0505 | 0.0645 | 0.0720 | 0.0666 | 0.0633 | 0.0485 | 0.0613 | 0.0591 | 0.0612 | 0.1345 | 0.1173 | 0.1319 |
| | Brier Score | 0.2543 | 0.6838 | 0.7566 | 0.9450 | 0.7421 | 0.0595 | 0.2003 | 0.1967 | 0.0549 | 0.0470 | 0.0301 | 0.0675 |
| | Accuracy | 0.6245 | 0.3040 | 0.1230 | 0.0470 | 0.2300 | 0.9480 | 0.8010 | 0.8090 | 0.9385 | 0.9505 | 0.9663 | 0.9210 |
| **Tasi_GUIDE_BS\*** | AUC ROC | 0.7528 | 0.7151 | 0.6508 | 0.7718 | 0.7952 | 0.7312 | 0.9412 | 0.9449 | **0.9964** | 0.9930 | 0.9713 | 0.9909 |
| | Precision | 0.5341 | 0.5367 | 0.4677 | 0.7023 | 0.6763 | 0.5653 | 0.8822 | 0.9138 | **0.9933** | 0.9861 | 0.9522 | 0.9861 |
| | Recall | 0.8784 | 0.9011 | 0.8842 | 0.9294 | 1.0000 | 0.4492 | 1.0000 | 1.0000 | **0.9746** | 0.9435 | 0.7680 | 0.8588 |
| | F1 Score | 0.6568 | 0.5411 | 0.5283 | 0.5595 | 0.5476 | 0.5282 | 0.5229 | 0.5229 | **0.9705** | 0.9612 | 0.8599 | 0.9129 |
| | Brier Score | 0.2213 | 0.5245 | 0.4841 | 0.5067 | 0.5733 | 0.2172 | 0.6457 | 0.6454 | **0.0182** | 0.0216 | 0.0840 | 0.0458 |
| | Accuracy | 0.6750 | 0.4590 | 0.4410 | 0.4820 | 0.4150 | 0.7160 | 0.3540 | 0.3540 | **0.9790** | 0.9730 | 0.9115 | 0.9420 |
| Listgarten_GUIDE_BS\* | AUC ROC | 0.7896 | 0.6300 | 0.5541 | 0.7792 | 0.7461 | 0.7317 | 0.8204 | 0.7975 | 0.9224 | **0.9395** | 0.7819 | 0.9340 |
| | Precision | 0.1261 | 0.0897 | 0.0986 | 0.1491 | 0.1573 | 0.1177 | 0.3388 | 0.2091 | 0.6376 | **0.6654** | 0.1773 | 0.6347 |
| | Recall | 0.9464 | 0.9643 | 1.0000 | 1.0000 | 1.0000 | 0.0000 | 1.0000 | 1.0000 | 0.5357 | **0.4286** | 0.0000 | 0.6786 |
| | F1 Score | 0.2179 | 0.1234 | 0.1067 | 0.1167 | 0.1163 | 0.0000 | 0.1061 | 0.1061 | 0.5646 | **0.5524** | 0.0000 | 0.5803 |
| | Brier Score | 0.2608 | 0.7585 | 0.8058 | 0.8393 | 0.8341 | 0.0681 | 0.9438 | 0.9437 | 0.0389 | **0.0381** | 0.0514 | 0.0422 |
| | Accuracy | 0.6195 | 0.2330 | 0.0625 | 0.1520 | 0.1490 | 0.9260 | 0.0560 | 0.0560 | 0.9537 | **0.9610** | 0.9440 | 0.9450 |
| Kleinstiver_GUIDE_BS\* | AUC ROC | 0.7602 | 0.5580 | 0.4803 | 0.7566 | 0.7649 | 0.6457 | 0.9377 | 0.9412 | **0.9690** | 0.9505 | 0.7497 | 0.9643 |
| | Precision | 0.1126 | 0.0708 | 0.0899 | 0.2345 | 0.2391 | 0.0980 | 0.6364 | 0.6774 | **0.6964** | 0.7314 | 0.1759 | 0.6776 |
| | Recall | 0.7222 | 0.8696 | 1.0000 | 1.0000 | 0.9815 | 0.0000 | 1.0000 | 1.0000 | **0.5741** | 0.6852 | 0.0000 | 0.5185 |
| | F1 Score | 0.1973 | 0.1137 | 0.1086 | 0.1070 | 0.1168 | 0.0000 | 0.1025 | 0.1025 | **0.6596** | 0.7048 | 0.0000 | 0.6222 |
| | Brier Score | 0.2136 | 0.7190 | 0.7624 | 0.8903 | 0.7784 | 0.0590 | 0.9458 | 0.9455 | **0.0276** | 0.0294 | 0.0496 | 0.0244 |
| | Accuracy | 0.6827 | 0.2727 | 0.1130 | 0.0990 | 0.1980 | 0.9420 | 0.0540 | 0.0540 | **0.9680** | 0.9690 | 0.9462 | 0.9660 |
| Hmg_BS | AUC ROC | 0.6815 | 0.7501 | 0.7508 | 0.5313 | 0.7387 | 0.5976 | 0.8736 | 0.8782 | 0.9419 | 0.9508 | 0.7417 | 0.8817 |
| | Precision | 0.0797 | 0.1109 | 0.1485 | 0.0529 | 0.1140 | 0.0722 | 0.3842 | 0.3497 | 0.6047 | 0.4935 | 0.2402 | 0.4825 |
| | Recall | 0.9807 | 1.0000 | 1.0000 | 1.0000 | 0.9808 | 0.0000 | 1.0000 | 1.0000 | 0.9615 | 0.8654 | 0.0000 | 0.9615 |
| | F1 Score | 0.1417 | 0.1034 | 0.0990 | 0.1034 | 0.1079 | 0.0000 | 0.0989 | 0.0989 | 0.2813 | 0.4823 | 0.0000 | 0.1741 |
| | Brier Score | 0.3939 | 0.8954 | 0.8136 | 0.8969 | 0.8278 | 0.0603 | 0.9479 | 0.9476 | 0.2098 | 0.0883 | 0.0475 | 0.3634 |
| | Accuracy | 0.3823 | 0.0980 | 0.0530 | 0.0985 | 0.1570 | 0.9435 | 0.0520 | 0.0520 | 0.7445 | 0.9027 | 0.9480 | 0.5253 |

In the third scenario, the SITE dataset served as the source dataset in our transfer learning experiments. The obtained ROC curves (see Fig 5C) and PR curves (see Fig A(C) in S1 Fig) are presented for all considered ML and DL models trained on the SITE dataset and evaluated on its bootstrapped counterpart, SITE_BS. Additionally, Fig 8 compares the ROC curves for all considered models, using the complete SITE dataset as source and the bootstrapped variants of the remaining datasets as targets (for PR curves see Fig D in S1 Fig). Further quantitative results are provided in Table 6. For the SITE_BS target dataset, FNN3 and MLP2 emerged as the best-performing models across all metrics. When the Kleinstiver_BS and Hmg_BS datasets were used as targets, the FNN5 model demonstrated notable results across diverse evaluation metrics.

Based on the evaluation results summarized in Tables 4, 5, and 6 across the three scenarios, the CD33, CIRCLE, and SITE datasets were found to be the most suitable sources for their respective bootstrapped counterparts: CD33_BS, CIRCLE_BS, and SITE_BS. This result

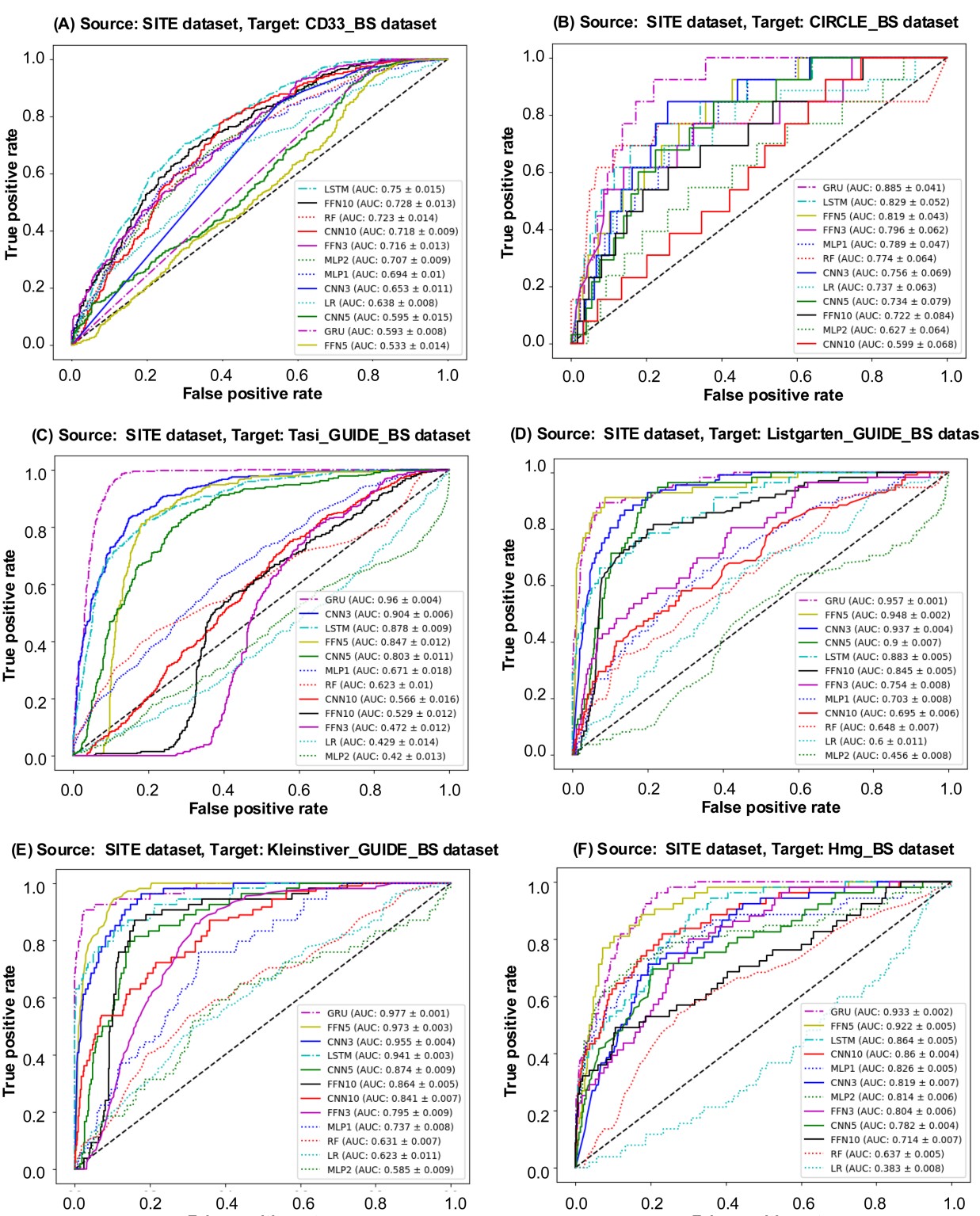

**Fig 8. ROC curves for model evaluation.** ROC curves for models trained on the SITE dataset, used as source, and six bootstrapped datasets: CD33_BS, CIRCLE_BS, Tasi_GUIDE_BS, Listgarten_GUIDE_BS, Kleinstiver_GUIDE_BS, Hmg_BS, used as targets. The AUC ROC values for each model are displayed in descending order within the figure.

**Table 6. Performance metrics for each considered classification model obtained using the SITE dataset for training (i.e. as source).** Target datasets exhibiting the highest similarity to the SITE dataset are marked with an asterisk. The results of the top-performing models are highlighted in bold.

| Target | Metric | FNN3 | FNN5 | FNN10 | CNN3 | CNN5 | CNN10 | LSTM | GRU | MLP1 | MLP2 | RF | LR |
|---|---|---|---|---|---|---|---|---|---|---|---|---|---|
| CD33_BS | AUC ROC | 0.7158 | 0.5332 | 0.7281 | 0.6531 | 0.5949 | 0.7178 | 0.7497 | 0.5926 | 0.6945 | 0.7074 | 0.7226 | 0.6376 |
| | Precision | 0.6802 | 0.4725 | 0.6852 | 0.5647 | 0.5598 | 0.6488 | 0.6777 | 0.5192 | 0.6619 | 0.6372 | 0.6631 | 0.6051 |
| | Recall | 0.5021 | 0.6282 | 0.7115 | 1.0000 | 1.0000 | 0.9808 | 1.0000 | 1.0000 | 0.4615 | 0.8568 | 0.0000 | 0.0000 |
| | F1 Score | 0.5725 | 0.5460 | 0.6755 | 0.6376 | 0.6376 | 0.6770 | 0.6376 | 0.6376 | 0.5461 | 0.6820 | 0.0000 | 0.0000 |
| | Brier Score | 0.2577 | 0.2694 | 0.2216 | 0.5318 | 0.5137 | 0.3577 | 0.5265 | 0.5320 | 0.3129 | 0.3294 | 0.4422 | 0.4665 |
| | Accuracy | 0.6490 | 0.5110 | 0.6800 | 0.4680 | 0.4680 | 0.5620 | 0.4680 | 0.4680 | 0.6410 | 0.6260 | 0.5320 | 0.5320 |
| CIRCLE_BS | AUC ROC | 0.7962 | 0.8191 | 0.7221 | 0.7563 | 0.7340 | 0.5987 | 0.8287 | 0.8854 | 0.7894 | 0.6273 | 0.7739 | 0.7371 |
| | Precision | 0.1440 | 0.1565 | 0.1077 | 0.1034 | 0.1188 | 0.0536 | 0.1704 | 0.2002 | 0.1270 | 0.0634 | 0.2160 | 0.1158 |
| | Recall | 0.0385 | 0.0220 | 0.0684 | 1.0000 | 0.8462 | 0.2462 | 0.9872 | 1.0000 | 0.0849 | 0.0559 | 0.0000 | 0.0000 |
| | F1 Score | 0.0571 | 0.0323 | 0.0757 | 0.0586 | 0.0854 | 0.0736 | 0.0627 | 0.0817 | 0.1008 | 0.0429 | 0.0000 | 0.0000 |
| | Brier Score | 0.0282 | 0.0280 | 0.0375 | 0.7098 | 0.3803 | 0.1235 | 0.5620 | 0.5001 | 0.0345 | 0.0577 | 0.0238 | 0.0255 |
| | Accuracy | 0.9670 | 0.9681 | 0.9570 | 0.1640 | 0.5292 | 0.8393 | 0.2330 | 0.4152 | 0.9627 | 0.9360 | 0.9740 | 0.9740 |
| **SITE_BS*** | AUC ROC | **0.9939** | 0.9883 | 0.9820 | 0.8019 | 0.7815 | 0.7767 | 0.8729 | 0.9142 | 0.9908 | **0.9881** | 0.9875 | 0.8795 |
| | Precision | **0.9373** | 0.8747 | 0.8838 | 0.1866 | 0.1367 | 0.1163 | 0.4825 | 0.5284 | 0.9547 | **0.9613** | 0.9338 | 0.5560 |
| | Recall | **0.6176** | 0.4706 | 0.4706 | 0.9706 | 0.9403 | 0.6863 | 1.0000 | 0.9118 | 0.8235 | **0.9113** | 0.8209 | 0.3262 |
| | F1 Score | **0.7565** | 0.6275 | 0.6275 | 0.0831 | 0.0882 | 0.1514 | 0.0799 | 0.1426 | 0.9032 | **0.9450** | 0.9016 | 0.4787 |
| | Brier Score | **0.0118** | 0.0144 | 0.0154 | 0.6250 | 0.5370 | 0.2173 | 0.5822 | 0.3077 | 0.0061 | **0.0035** | 0.0073 | 0.0208 |
| | Accuracy | **0.9865** | 0.9810 | 0.9810 | 0.2710 | 0.3432 | 0.7397 | 0.2170 | 0.6270 | 0.9940 | **0.9965** | 0.9940 | 0.9768 |
| Tasi_GUIDE_BS | AUC ROC | 0.4724 | 0.8475 | 0.5287 | 0.9041 | 0.8027 | 0.5658 | 0.8784 | 0.9597 | 0.6712 | 0.4202 | 0.6226 | 0.4288 |
| | Precision | 0.3086 | 0.6122 | 0.3374 | 0.8173 | 0.6037 | 0.3693 | 0.7931 | 0.9072 | 0.5243 | 0.3196 | 0.5215 | 0.3138 |
| | Recall | 0.0932 | 0.1723 | 0.1384 | 1.0000 | 0.9520 | 0.2740 | 0.9972 | 1.0000 | 0.1461 | 0.1864 | 0.0565 | 0.0000 |
| | F1 Score | 0.1048 | 0.2552 | 0.1620 | 0.5527 | 0.6172 | 0.3201 | 0.5700 | 0.6093 | 0.2381 | 0.2333 | 0.1067 | 0.0000 |
| | Brier Score | 0.5417 | 0.2923 | 0.4474 | 0.4799 | 0.3151 | 0.3362 | 0.4143 | 0.3880 | 0.3131 | 0.4184 | 0.3067 | 0.3477 |
| | Accuracy | 0.4360 | 0.6440 | 0.4930 | 0.4270 | 0.5820 | 0.5880 | 0.4675 | 0.5460 | 0.6693 | 0.5660 | 0.6650 | 0.6460 |
| Listgarten_GUIDE_BS | AUC ROC | 0.7540 | 0.9485 | 0.8447 | 0.9366 | 0.9003 | 0.6951 | 0.8826 | 0.9570 | 0.7035 | 0.4565 | 0.6475 | 0.6002 |
| | Precision | 0.1793 | 0.6272 | 0.2161 | 0.4875 | 0.2725 | 0.1548 | 0.5412 | 0.7098 | 0.1302 | 0.0521 | 0.1279 | 0.0775 |
| | Recall | 0.0893 | 0.4286 | 0.1429 | 1.0000 | 1.0000 | 0.3214 | 1.0000 | 1.0000 | 0.1071 | 0.0893 | 0.0000 | 0.0000 |
| | F1 Score | 0.1389 | 0.5393 | 0.1517 | 0.1208 | 0.1528 | 0.2130 | 0.1109 | 0.1615 | 0.1270 | 0.0400 | 0.0000 | 0.0000 |
| | Brier Score | 0.0592 | 0.0303 | 0.0684 | 0.7149 | 0.4775 | 0.0955 | 0.7056 | 0.4989 | 0.0740 | 0.2192 | 0.0524 | 0.0566 |
| | Accuracy | 0.9380 | 0.9590 | 0.9103 | 0.1845 | 0.3790 | 0.8670 | 0.1020 | 0.4185 | 0.9175 | 0.7600 | 0.9440 | 0.9420 |
| **Kleinstiver_GUIDE_BS*** | AUC ROC | 0.7946 | **0.9726** | 0.8640 | 0.9547 | 0.8736 | 0.8406 | 0.9409 | 0.9769 | 0.7366 | 0.5852 | 0.6313 | 0.6227 |
| | Precision | 0.1231 | **0.6514** | 0.1887 | 0.6502 | 0.2419 | 0.3640 | 0.7339 | 0.8588 | 0.1246 | 0.0780 | 0.1111 | 0.1128 |
| | Recall | 0.0556 | **0.5001** | 0.2407 | 1.0000 | 0.9815 | 0.6111 | 1.0000 | 1.0000 | 0.0741 | 0.1667 | 0.0000 | 0.0000 |
| | F1 Score | 0.0619 | **0.5902** | 0.1745 | 0.1155 | 0.1608 | 0.3002 | 0.1425 | 0.2097 | 0.0879 | 0.1395 | 0.0000 | 0.0000 |
| | Brier Score | 0.0831 | **0.0286** | 0.0860 | 0.7231 | 0.4254 | 0.0994 | 0.4717 | 0.3086 | 0.0709 | 0.1052 | 0.0507 | 0.0529 |
| | Accuracy | 0.9086 | **0.9625** | 0.8770 | 0.1730 | 0.4470 | 0.8460 | 0.3500 | 0.5930 | 0.9170 | 0.8890 | 0.9460 | 0.9457 |
| **Hmg_BS*** | AUC ROC | 0.8035 | **0.9225** | 0.7144 | 0.8187 | 0.7817 | 0.8598 | 0.8642 | 0.9326 | 0.8259 | 0.8140 | 0.6368 | 0.3828 |
| | Precision | 0.3102 | **0.4260** | 0.3091 | 0.1840 | 0.2073 | 0.3882 | 0.4257 | 0.5081 | 0.3882 | 0.3666 | 0.0810 | 0.0396 |
| | Recall | 0.1123 | **0.7692** | 0.3058 | 1.0000 | 0.9805 | 0.4423 | 1.0000 | 1.0000 | 0.2692 | 0.4423 | 0.0000 | 0.0000 |
| | F1 Score | 0.1941 | **0.5011** | 0.3795 | 0.1042 | 0.1145 | 0.3566 | 0.1144 | 0.1244 | 0.3590 | 0.4600 | 0.0000 | 0.0000 |
| | Brier Score | 0.0437 | **0.0599** | 0.0481 | 0.8443 | 0.6543 | 0.0587 | 0.6440 | 0.6359 | 0.0432 | 0.0502 | 0.0491 | 0.0520 |
| | Accuracy | 0.9518 | **0.9204** | 0.9485 | 0.1060 | 0.2217 | 0.9170 | 0.1950 | 0.2680 | 0.9500 | 0.9460 | 0.9480 | 0.9480 |

was rather expected given the highest similarity scores between the complete datasets and their bootstrapped counterparts observed for all the three similarity measures considered (see Table 3). Furthermore, among the three source datasets (CD33, SITE, and CIRCLE), CIRCLE was identified as the optimal source for the Tasi_GUIDE_BS target dataset across all metrics, when using the MLP1 model (this corresponds to the highest similarity scores between Tasi_GUIDE_BS and CIRCLE provided by cosine, Euclidean, and Manhattan metrics; see Table 3), and for the Listgarten_GUIDE_BS target dataset, also across all metrics, when using the MLP2 model (this corresponds to the highest similarity score between Listgarten_GUIDE_BS and CIRCLE provided by cosine similarity; see Table 3). When the Kleinstiver_GUIDE_BS dataset was used as target, the CIRCLE (with the MLP1 and MLP2 models) and SITE (with the FNN5 model) datasets emerged as the optimal sources (see Tables 5 and 6). This corresponds to the highest similarity score between Kleinstiver_GUIDE_BS and both

CIRCLE and SITE provided by cosine similarity (see Table 3). When the Hmg_BS dataset was used as target, the SITE dataset (using the FNN5 model) was identified as the optimal source (see Table 6); once again, this reflects the highest similarity score between Hmg_BS and SITE provided by cosine similarity (see Table 3).

These results validate two critical points of our study: First, similarity score results are reliable and trustworthy indices for determining the most appropriate source dataset for a given target dataset prior to performing transfer learning experiments in CRISPR-Cas9. They reinforce the effectiveness of our methodology as a robust pre-selection tool for transfer learning, providing a systematic approach for identifying efficiently suitable source data. Second, cosine distance (or cosine similarity) emerges as the most dependable metric, among the three metrics considered, for selecting the most appropriate source dataset for transfer learning.

Moreover, our results clearly demonstrate that similarity-based source data pre-selection is necessary to mitigate negative knowledge transfers. If a source dataset is chosen solely by size, availability, or even class imbalance ratio, but without similarity assessment, this could eventually lead to a suboptimal or negative transfer. For example, the CIRCLE and SITE datasets considered in our study have comparable sizes and almost identical class imbalance ratios (0.0128 and 0.0176, respectively - see Table 1), but a knowledge transfer from CIRCLE to SITE as well as that from SITE to CIRCLE are clearly negative with the highest F1-score values of 0.1345 (see Table 5) and 0.1008 (see Table 6), respectively, over all competing ML and DL models. Our similarity-based analysis suggests that such transfers should be avoided (see Table 3).

It is worth noting that in some cases a potential source dataset with a slightly lower cosine similarity with the target might still yield competitive or even superior performance for specific models - MLP-based models in our case. This could be due to such factors as a richer representation of specific rare patterns important for the target task or some MLP inductive biases aligning better with the source data distribution. For example, the recommended transfers with lower cosine similarity scores of 0.5701 from CIRCLE to Listgarten_GUIDE_BS and of 0.5672 from CIRCLE to Kleinstiver_GUIDE_BS led to competitive knowledge transfers with the corresponding best F1-score score values of 0.5646 and 0.7048, obtained, respectively, with MLP1 and MLP2 (see Table 5). It was not so for Euclidean and Manhattan similarities whose highest values in this case led to transfers from CD33 (instead of CIRCLE) with much lower best F1-score score values of 0.2500 for Listgarten_GUIDE_BS and of 0.1946 for Kleinstiver_GUIDE_BS, both obtained with MLP1 (see Table 4).

## 3. Conclusion

This study explores the effectiveness and applicability of transfer learning in improving CRISPR-Cas9 off-target predictions by adapting a similarity-based approach. We consider three popular distance measures - cosine, Euclidean, and Manhattan distances to assess similarity between a given target dataset and an ensemble of potential source datasets. A candidate source dataset having the highest similarity with the given target can then be recommended for transfer learning experiments.

Establishing the most appropriate source dataset for a given target dataset in the transfer learning prospective is a relevant theoretical problem in itself. We show how it can be effectively solved in practice in the context of CRISPR-Cas9 off-target prediction. The main novelty of our study consists in the proposed similarity-based pre-evaluation rather than in an innovative transfer learning algorithm or an effective deep learning network architecture.

Our experiments were conducted using seven real-world CRISPR-Cas9 off-target datasets: CD33, CIRCLE, SITE, Tasi_GUIDE, Listgarten_GUIDE, Kleinstiver_GUIDE, and Hmg. The performance of various deep learning network architectures, i.e. CNNs, FNNs, LSTM-RNNs, GRU-RNNs, and MLPs, alongside two traditional machine learning models, i.e. RF and LR, was evaluated in a comprehensive simulation study. Six evaluation metrics, including AUC ROC, Precision, Recall, F1-score, Brier score, and Accuracy were considered. AUC ROC, F1-score, and Brier score are well adapted for assessing the model performances in our case since real-world CRISPR-Cas9 data are often highly imbalanced.

Our results indicate that cosine distance stands out as the most reliable and consistent measure for assessing similarity between two CRISPR-Cas9 datasets in terms of off-target transfer learning experiments. High similarity values provided by cosine similarity usually correspond to the top results achieved by the considered evaluation metrics. This was not always the case of Euclidean and Manhattan distances whose results were highly correlated as we worked with binary data representations. Overall, MLP variants 1 and 2, 3- and 5-layer FNNs, and an RNN-GRU turned out to be the best-performing models in our transfer learning scenarios. While these models tend to offer a superior performance in most cases, the choice between machine learning and deep learning models should depend on the characteristics of the given target and source datasets, taking into account the dataset sizes and an eventual class imbalance. The fact that in many instances two simple MLP models outperformed much more sophisticated RNN-GRU and, especially, RNN-LSTM neural network architectures is not very surprising since MLPs usually cope well with tabular data, such as our CRISPR-Cas9 one-hot encoded sequence datasets, allowing for capturing complex, non-linear patterns, whereas RNN-based models excel at capturing time-series patterns and long-term dependencies in complex scenarios, being particularly useful in natural language processing and speech recognition.

Our findings highlight the critical role of similarity-based insights in optimizing transfer learning workflows. Broader impacts of the proposed dual-layered framework are the following: (1) The new framework streamlines the transfer learning process by reducing the number of potential source datasets and recommended ML and DL models, and thus the number of trial-and-error attempts, which are convenient for a selected target dataset, and (2) it enables faster development of transfer learning models for CRISPR-Cas9 off-target prediction, which can now be successfully tested on mutually compatible sets (i.e. those with high cosine similarity scores) of source and target data.

In the future, we plan to compare our approach with transformer-based models optimized for tabular data [52] as well as with different data augmentation techniques allowing for better leverage of limited datasets [53]. Moreover, it would be interesting to extend the proposed similarity framework beyond sequence similarity by incorporating into it available biological and experimental information, such as species data, cell type, enzyme type, experimental conditions/technology being used, and data size. Specifically, each of these factors could be added to the discussed $7L$ input vectors as an extra component, normalized to [0,1] range for numerical data and one-hot encoded for categorical data. It would also be interesting to integrate the proposed similarity-based selection with deep learning architectures for domain adaptation that explicitly address distribution shifts between source and target [54,55]. In this case, the source labeled sample weights could ideally be calculated leveraging both distribution and similarity patterns of the source and target samples.

## Supporting information

**S1 Text. Detailed description of the considered ML and DL models and their hyperparameters.**
(PDF)

**S1 Table. Tables reporting hyperparameters of the considered ML and DL models.**
(PDF)

**S1 Fig. Precision-recall curves for three source datasets.**
(PDF)

## Acknowledgments

The authors thank Dr. Robert Nadon (McGill University) for his valuable comments on this manuscript.

## Author contributions

**Conceptualization:** Jeremy Charlier, Zeinab Sherkatghanad, Vladimir Makarenkov.

**Data curation:** Jeremy Charlier, Zeinab Sherkatghanad.

**Formal analysis:** Jeremy Charlier, Zeinab Sherkatghanad, Vladimir Makarenkov.

**Funding acquisition:** Vladimir Makarenkov.

**Investigation:** Zeinab Sherkatghanad, Vladimir Makarenkov.

**Methodology:** Jeremy Charlier, Zeinab Sherkatghanad, Vladimir Makarenkov.

**Project administration:** Vladimir Makarenkov.

**Software:** Jeremy Charlier, Zeinab Sherkatghanad.

**Supervision:** Vladimir Makarenkov.

**Validation:** Jeremy Charlier, Zeinab Sherkatghanad.

**Writing – original draft:** Jeremy Charlier, Zeinab Sherkatghanad, Vladimir Makarenkov.

**Writing – review & editing:** Jeremy Charlier, Zeinab Sherkatghanad, Vladimir Makarenkov.

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
