## [Decision Letter · Decision Letter 0]

25 Jun 2025

PCOMPBIOL-D-25-01030

Similarity-based transfer learning with deep learning networks for accurate CRISPR-Cas9 off-target prediction

PLOS Computational Biology

Dear Dr. Makarenkov,

Thank you for submitting your manuscript to PLOS Computational Biology. After careful consideration, we feel that it has merit but does not fully meet PLOS Computational Biology's publication criteria as it currently stands. Therefore, we invite you to submit a revised version of the manuscript that addresses the points raised during the review process.

Please submit your revised manuscript within 60 days Aug 25 2025 11:59PM. If you will need more time than this to complete your revisions, please reply to this message or contact the journal office at ploscompbiol@plos.org. Please include the following items when submitting your revised manuscript:

We look forward to receiving your revised manuscript.

Kind regards,

Lun Hu

Academic Editor

PLOS Computational Biology

Ilya Ioshikhes

Section Editor

PLOS Computational Biology

**Journal Requirements:**

At this stage, the following Authors/Authors require contributions: Jeremy Charlier, Zeinab Sherkatghanad, and Vladimir Makarenkov. Please ensure that the full contributions of each author are acknowledged in the "Add/Edit/Remove Authors" section of our submission form.

4) We have noticed that you have uploaded Supporting Information files, but you have not included a complete list of legends. Please add a full list of legends for your Supporting Information files after the references list.

Potential Copyright Issues:

i) Figures 1B, and 3. Please confirm whether you drew the images / clip-art within the figure panels by hand. If you did not draw the images, please provide (a) a link to the source of the images or icons and their license / terms of use; or (b) written permission from the copyright holder to publish the images or icons under our CC BY 4.0 license. Alternatively, you may replace the images with open source alternatives. See these open source resources you may use to replace images / clip-art:

6) Please provide a detailed Financial Disclosure statement. This is published with the article. It must therefore be completed in full sentences and contain the exact wording you wish to be published.

1) Please clarify all sources of financial support for your study. List the grants, grant numbers, and organizations that funded your study, including funding received from your institution. Please note that suppliers of material support, including research materials, should be recognized in the Acknowledgements section rather than in the Financial Disclosure

2) State the initials, alongside each funding source, of each author to receive each grant. For example: "This work was supported by the National Institutes of Health (####### to AM; ###### to CJ) and the National Science Foundation (###### to AM)."

3) State what role the funders took in the study. If the funders had no role in your study, please state: "The funders had no role in study design, data collection and analysis, decision to publish, or preparation of the manuscript."

4) If any authors received a salary from any of your funders, please state which authors and which funders..

7)  Please ensure that the funders and grant numbers match between the Financial Disclosure field and the Funding Information tab in your submission form. Note that the funders must be provided in the same order in both places as well.  

**Reviewers' comments:**

Reviewer's Responses to Questions

**Comments to the Authors:**

Reviewer #1: This manuscript focuses on showing the impact of transfer learning on CRISPRCas9 off-target identification. The number of used similarity metrics are enough. The used datasets are comprehensive. The problem has been extensively studied before, but transfer learning is the main novelty here. Even though the manuscrupt is written quite well, I have the following comments:

Major Comments:

1- Why did we use 7 in matrix? It is not clear to me as discussed at the top of page 11.

2- Can you explain in a bit more detail why MLP suprisingly performs better than sequence-based models such as GRU?

Minor Comments:

1- Can you mathematically describe the evaluation metrics?

2- The number of provided references are enough. But, in terms of transfer learning and transformers on financial domain, can you cite the following paper as well?

a- Gezici, A.H.B. and Sefer, E., 2024. Deep transformer-based asset price and direction prediction. IEEE Access, 12, pp.24164-24178

Reviewer #2: 1. Strengthen Introduction Regarding Algorithmic Novelty:

While you effectively highlight the challenge of data limitations and the benefit of transfer learning, more explicitly state that existing transfer learning applications in CRISPR-Cas9 (e.g., DeepCRISTL, C-RNNCrispr) often lack a principled method for source dataset selection. This will better position your similarity-based pre-evaluation as the key innovation, rather than claiming novelty in the transfer learning algorithms themselves. Clarify that your contribution is in optimizing the transfer learning process through intelligent source selection, not in inventing new deep learning architectures for transfer.

2. Elaborate on Data Representation and Distance Metrics (Methods):

Theoretical Basis for Distance Metrics: In Section 1.7, beyond presenting the formulas, delve deeper into the theoretical reasons why cosine, Euclidean, and Manhattan distances might behave differently with your specific 7×L encoded sgRNA-DNA sequence pairs, which are flattened into 7L vectors. Discuss how each metric responds to features like point-wise mismatches, insertions, or deletions given your encoding strategy (e.g., cosine's sensitivity to direction in high-dimensional sparse spaces, Euclidean's sensitivity to magnitude, Manhattan's robustness to outliers). This will provide a more sophisticated understanding of why cosine distance ultimately performs best.

Normalization Details: Clearly specify the exact Min-Max normalization formula used for "1-normalized_average_distance" in Section 1.7 or a new "Data Processing" subsection. Explicitly state whether this normalization is applied per distance metric independently or across all metrics collectively. This is crucial for reproducibility and transparent interpretation of "similarity values" in Table 3.

3. Detailed Justification of Sampling and Bootstrapping Strategies (Methods):

Provide a clearer rationale for choosing a bootstrapped target dataset size of 250. Is this based on empirical observation of minimum samples needed for stable predictions, or a specific computational constraint? Similarly, justify the choice of n=5,000 iterations for random sampling within source datasets for distance calculation. Explain how these choices balance computational feasibility with the statistical representativeness of the similarity assessment, particularly in relation to the original large datasets.

4. In-depth Discussion of Similarity-Performance Link (Results and Discussion):

Interpreting Metric Discrepancies: While you note that Euclidean and Manhattan distances provide larger differences between recommended and non-recommended sources but cosine gives higher overall similarity, delve deeper into why this occurs. For instance, do Euclidean/Manhattan distances, being sensitive to absolute differences, better highlight stark dissimilarity, even if overall magnitude is less important for transferability than feature direction (captured by cosine)? Connect these observations more directly to the underlying biological or sequence characteristics.

Illustrating Negative Transfer Avoidance: Explicitly discuss how your similarity-based pre-selection strategy mitigates negative transfer. Can you point to instances where a source dataset, if chosen solely by size or availability (without similarity assessment), would have led to suboptimal or negative transfer, and how your method successfully avoided this? Providing even a hypothetical or qualitative example could strengthen this point.

Performance Beyond Similarity: While you effectively demonstrate that high cosine similarity often correlates with top model performance, also discuss cases where a source dataset with slightly lower cosine similarity might still yield competitive or even superior performance for specific models due to other factors (e.g., a richer representation of specific rare patterns important for the target task, or the specific model's inductive biases aligning better with that particular source's data distribution). This would add nuance to the "reliability" claim.

5. Refine Conclusion and Future Work:

Concrete Future Directions: Beyond proposing "more advanced similarity measures" or "adaptive transfer learning strategies", offer concrete examples. For instance, could your similarity framework be extended to incorporate biological contextual information (e.g., cell type, experimental conditions) beyond just sequence similarity? Could new deep learning architectures for domain adaptation (which explicitly address distribution shifts between source and target) be integrated with your similarity-based selection?

Broader Impact: Elaborate on the broader impact of your dual-layered framework. How specifically does it "streamline the transfer learning process"? Does it reduce trial-and-error, save computational resources, or enable faster development of CRISPR-Cas9 off-target prediction models for new experimental data?

6. Minor Revisions:

Figure Titles and Labels: Ensure consistency between figure titles and the text. For example, Figure 4's subplot titles refer to "distance" (e.g., "Cosine distance for CD33 dataset") , while the main text and Table 3 refer to "similarity" (which is 1 - normalized distance). Please unify this terminology to avoid confusion. Suggestion: "Cosine Similarity (1 - Normalized Distance) for CD33 dataset".

Reviewer #3: The work has written well and has significance merits but there are minor changes are required for acceptance.

1. Clarify what the term “discriminator score” refers to in the dual anomaly scoring mentioned in the abstract.

2. Rewrite the opening sentence of the introduction for brevity and improved clarity.

3. Reformat the literature review section into bullets or a table to clearly highlight key study comparisons.

4. Ensure consistent usage of the term “self-adaptive” throughout the manuscript.

5. Revise lengthy sentences for better grammar and readability, particularly in the introduction.

6. Add a summary paragraph highlighting similarities and differences among reviewed studies to contextualize SAD-GAN’s contribution.

7. Define and explain all mathematical symbols (α, β, G, D) at their first mention for clarity.

8. Elaborate on how alerts are triggered and communicated during anomaly detection.

9. Justify the use of non-overlapping time windows in Section 3.4 by comparing with alternatives like sliding windows.

10. Provide details on how static and dynamic thresholds are implemented in Section 4.5.

11. Clarify how the system enables autonomous anomaly detection and response without prior planning.

12. Include a discussion on the limitations or drawbacks of the proposed approach.

13. Expand the future work section by outlining 4–5 specific research directions.

**Have the authors made all data and (if applicable) computational code underlying the findings in their manuscript fully available?**

Reviewer #1: Yes

Reviewer #2: None

Reviewer #3: Yes

PLOS authors have the option to publish the peer review history of their article (what does this mean?). If published, this will include your full peer review and any attached files.

Reviewer #1: **Yes: **Emre Sefer

Reviewer #2: No

Reviewer #3: **Yes: **surjeet dalal

**Figure resubmission:**
---

## [Decision Letter · Decision Letter 1]

9 Oct 2025

Dear Dr. Makarenkov,

We are pleased to inform you that your manuscript 'Similarity-based transfer learning with deep learning networks for accurate CRISPR-Cas9 off-target prediction' has been provisionally accepted for publication in PLOS Computational Biology.

Best regards,

Lun Hu

Academic Editor

PLOS Computational Biology

Ilya Ioshikhes

Section Editor

PLOS Computational Biology

All reviewers were satisified with the changes made in this revision, and they have no further comments.

Reviewer's Responses to Questions

**Comments to the Authors:**

Reviewer #1: The authors have addressed all my comments, so I accept

Reviewer #2: The authors have carefully addressed all my previous comments. The revisions are adequate, and the additional explanations significantly improve the clarity and rigor of the manuscript. I believe the paper is now suitable for publication in its current form.

**Have the authors made all data and (if applicable) computational code underlying the findings in their manuscript fully available?**

Reviewer #1: Yes

Reviewer #2: Yes

PLOS authors have the option to publish the peer review history of their article (what does this mean?). If published, this will include your full peer review and any attached files.

Reviewer #1: **Yes: **Emre Sefer

Reviewer #2: No

---

## [Editor Report · Acceptance letter]

PCOMPBIOL-D-25-01030R1

Similarity-based transfer learning with deep learning networks for accurate CRISPR-Cas9 off-target prediction

Dear Dr Makarenkov,

I am pleased to inform you that your manuscript has been formally accepted for publication in PLOS Computational Biology. Your manuscript is now with our production department and you will be notified of the publication date in due course.

With kind regards,

Zsofia Freund
